

# Application of Parametric Speakers to

# Radio Acoustic Sounding System

by

Ahoro ADACHI[1] and Hiroyuki HASHIGUCHI[2]

[1]Meteorological Research Institute, Japan Meteorological Agency

1-1 Nagamine, Tsukuba, Ibaraki 305-0052, Japan

[2] Research Institute for Sustainable Humanosphere, Kyoto University

Gokasho, Uji, Kyoto 611-0011, Japan

*Correspondence to*: Ahoro ADACHI (aadachi@mri-jma.go.jp)

Tel: +81-29-853-8584 / Fax: +81-29-856-0644

Paper re-submitted on 25 September 2019 to:

Atmospheric Measurement Techniques

**Abstract**
In this study, a wind profiler with radio acoustic sounding system (RASS) and
operational radiosonde measurements were used to investigate the technical
practicability and reliability of using parametric speakers to measure the vertical profile
of virtual temperature. Characteristics of parametric speakers include high directivity
and very low sidelobes, which are preferable for RASS, especially those operating at
urban areas. The experiments were conducted on fine days with light winds to mitigate
the effects of the horizontal and vertical components of wind on acoustic waves used for
RASS. The results of this study indicated that, although parametric speaker RASS is
susceptible to horizontal winds due to the narrower acoustic beam, bias and standard
deviation of parametric speaker RASS versus radiosonde virtual temperature difference
(0.1°C, 0.4°C) were close to that from acoustic speakers (0.0°C, 0.4°C). In addition,
when compared with acoustic speaker RASS, the values for the parametric speaker
RASS were even smaller (0.1°C, 0.2°C). Based on these results, it is concluded that the
parametric speaker RASS has accuracy and precision comparable with acoustic speaker
RASS despite its high directivity of sound.

**1 Introduction**

Accurate measurements of temperature are essential in weather forecasting and studies

of atmospheric dynamics at all scales. The radio acoustic sounding system (RASS) is a

ground-based remote sensing technique that provides vertical profiles of virtual

temperature from a few hundred meters above the surface up to several kilometers in

elevation (Marshall et al., 1972; Peters et al., 1985). RASS technique has been applied

to wind profilers, whereby vertical profiles of virtual temperature can be measured with

the same temporal and spatial resolution that the profiler uses to measure winds (e.g.,

Adachi et al., 2005) with a relatively high degree of reliability (Matuura et al., 1986;

Moran et al., 1991; Angevine and Ecklund, 1994).

When using RASS techniques, one or more acoustic sources are co-located with

an antenna, and the profiler provides the vertical profile of the speed at which the

acoustic disturbance propagates vertically (Angevine et al., 1994). RASS temperature

measurements can be obtained on the basis of the relationship between the virtual

temperature $T_v$ (°C), the local speed of sound $C_a$ (m s$^{-1}$) and the measured radial wind

speed $w$ (m s$^{-1}$), and a good approximation can be obtained by

$$\qquad\qquad T_v = \left(\frac{c_a - w}{20.047}\right)^2 - 273.15. \qquad\qquad\qquad (1)$$
Thus, a vertical profile of the speed of sound can be converted to a profile of virtual
temperature. The radial wind speed is considered in Eq. (1) because the neglect of the
wind velocity along the beam may be a largest source of error in RASS measurements
(e.g., May et al., 1989; Angevine et al., 1994). However, we could not consider the
radial wind speed in our experiments, because strong clutter sometimes contaminated
the Doppler spectrum and masked the atmospheric echo in the vertical beam
observation. This issue is addressed in later sections.
The systematic error or bias of the virtual temperature measurements from RASS
observations have been shown to be less than 1°C, while the standard deviation or
precision has also been reported around 1°C. May et al. (1989) compared virtual
temperatures obtained from 915 and 50 MHz RASS with those obtained from
radiosonde measurements. The RASS data was averaged over approximately 6 min, and
about 50 soundings covering both the summer and winter seasons were examined. Both
the bias and the standard deviation were about 1°C, even without the application of the
vertical velocity correction. On the other hand, Martner et al. (1993) assessed the
performance of 915, 404 and 50 MHz wind profilers with RASS by comparing with
about 150 radiosonde measurements. They found that the bias (standard deviation) was
less than 0.3°C (about 1°C) for most systems, even though they did not make the
correction for vertical air motions, as the comparison was made under low vertical wind
conditions. Moran and Strauch (1994) compared temperature profiles obtained using a
VHF wind profiler with RASS with those obtained from radiosondes during a 5-week
period. They reported that the accuracy (standard deviation) was 0.9°C (less than 1°C),
after the application of the vertical velocity correction. Moreover, Angevine et al.
(1998) compared the virtual temperature measured by a 915 MHz wind profiler with
RASS with *in situ* observations at 396 m AGL on a tower. They found that the precision
of the RASS measurements was less than 0.9 K after the application of the vertical
velocity correction and corrections for thermodynamic constants. In addition, Görsdorf
and Lehmann (2000) reported that the bias (standard deviation) of the RASS
measurements with a 1.3 GHz wind profiler is 0.1K (0.7K) from the data observed for a
year compared with radiosondes if accurate corrections for vertical velocity, range, and
thermodynamic constants were applied. On the other hand, the height coverage of
RASS depends on the radio wave frequency deployed (May et al., 1988; Martner et al.,
1993) but is also limited by both the advection of the sound wave with the horizontal
wind and the atmospheric attenuation of the acoustic signal in addition to the effects of
turbulence and vertical temperature gradients (Lataitis, 1992).
A wind profiler with RASS has been frequently used to study the dynamics of the
atmosphere, especially in the boundary layer (e.g., Neiman et al., 1992; Peters and
Kirtzel, 1994; May, 1999; Bianco and Wilczak, 2002; White et al., 2003; Adachi et al.,
2004; Hashiguchi et al., 2004; Chandrasekhar Sarma et al., 2008). Among the
limitations of this method, an important one is the emission of strong sound waves,
whose frequency cannot be arbitrarily selected, but determined by the wavelength of the
radio wave used by the profiler to match the Bragg condition (the acoustic wavelength
$\lambda_a$ is equal to half the electromagnetic wavelength $\lambda_e$). Although the acoustic speakers
used for RASS measurements are usually co-located with the antenna and directed
vertically so that the generated sound wave propagates along the radio wave, a large
portion of the sound wave leaks horizontally because of the sidelobes of the speakers,
which prevents the temporal and/or continuous operation of RASS in urban
environments (Wulfmeyer et al., 2015). Thus, a new type of speaker that has extremely
low sidelobes would be ideal for RASS measurements.
A theoretical study of parametric speakers (or parametric acoustic array, PAA)
was established by Westervelt (1963). That study revealed that the nonlinear interaction
between two collimated high-frequency sound beams in an ideal fluid medium produces
two new waves with a sum and difference frequencies, and the latter may be used to
produce narrow beams of sound at relatively low frequencies in the audible range.
Berktay and Leahy (1974) presented a theoretical description that can be used to
compute the far field response of a parametric array for multiple sets of parameters.
Thereafter, the use of parametric arrays underwater has been the subject of a number of
theoretical and experimental studies. On the other hand, an experimental investigation
of the parametric array in air was first demonstrated by Bennett and Blackstock (1975),
and recently, the parametric loud speaker has become available for audio and speech
applications (Gan et al., 2012). The properties of parametric speakers include high
directivity and very low sidelobes, which are preferable for RASS measurements.
However, to the best of our knowledge, there are few, if any, studies on RASS
techniques using this type of speaker.

In this study, a detailed evaluation of the parametric speaker for RASS

measurements was conducted by comparing temperature data derived from this type of
speaker and those from both radiosonde and acoustic speaker RASS at the
Meteorological Research Institute (MRI) field site in Tsukuba, Japan. Instrumentation
and data analysis techniques are presented in Section 2. Results are presented in Section
3 and discussed in Section 4. Finally, a summary of our conclusions are presented in
Section 5.

**2 Instrumentation and data analysis techniques**

The MRI wind profiler, a four-panel LAP-3000 with RASS (Fig. 1a), is the type

originally developed at the National Oceanic and Atmospheric Administration (NOAA)
Aeronomy Laboratory (Carter et al., 1995; Ecklund et al., 1988). The profiler used in
this study operated at 1357.5 MHz with 100 m pulse lengths and a minimum
(maximum) gate of 200 m (1300 m) from the antenna in RASS mode. The vertical
resolution was set to 100 m based on the requirements for the Global Climate Observing
System (GCOS) Reference Upper-Air Network (GRUAN) by the WMO (2007). The
effect of the vertical air motion was not considered for RASS measurements in the
experiments because strong clutter caused by automobiles on a nearby highway
sometimes contaminated the Doppler spectrum and masked the atmospheric echo
(Adachi et al., 2004).
The configuration and operating parameters of the wind profiler with RASS are
summarized in Table 1. The antenna of the profiler was co-located with four acoustic
speakers in cylindrical enclosures and a parametric speaker, which was mounted on top
of a shed (Fig. 1a). For the experiment purpose, the RASS measurements were made
continuously for about an hour without wind observations. Since the wind profiler
operated at 1.3 GHz, the frequency of the acoustic source for the RASS measurement
was set at about 3 kHz to match the Bragg condition. Prior to every experiment, an
acoustic wave with a wide frequency range (2715 to 3265 Hz corresponding to about
±50°C) was generated to detect center Doppler frequency of the RASS echo. Then,
during each experiment, the emitted acoustic frequency range was automatically
narrowed down to a shorter frequency span (130 Hz, corresponding to about ±12°C)
around the detected center frequency to increase SNR and height coverage. The
frequency sweeps were randomly shuffled within each frequency range to make
acoustic spectrum almost uniform (Angevine et al., 1994).

The MRI parametric speaker, 100FM-001, consists of an array of more than

10,000 piezoelectric ceramic transducers configured on a semi-circular board with a
diameter of 1.8 m (Fig. 1b). The transducers were divided into 278 segments, with each
one mounted on the hexagonal board (Fig. 1c). The FPGA modules in the speaker
system were used to control the phase of the signals fed into the segments to generate
the acoustic beam with a particular preferred width and direction like other PAAs (e.g.,
Wu et al., 2012). The configuration and operating parameters of the speaker are
summarized in Table 2.

One of the desirable features of the PAA for RASS measurements is high

directivity of the sound beam. Prior to the designing of the MRI PAA, we made a
preliminary field sensitivity test for RASS using a prototype PAA with a beam width
smaller than 2° and relatively small power, but no RASS echo was observed. We
modified the prototype to broaden the beam width to about 6° or more, and the RASS
echo was observed up to a few range gates. We concluded that too narrow a beam is not
good for the RASS observation, and the PAA beam width should match that of the
profiler radio wave. Because the beam width of the MRI profiler is less than 7° (Table
1), the default sound beam width of the speaker was designed to be 5° (Table 2).
Although the latter width is somewhat smaller than that of the former, the RASS focal
spot determined by the sound beam width may be broadened by turbulence (Lataitis,
1992) and match the radio beam width, which is preferable for RASS measurements.

In order to measure the audible sound pressure level (SPL) pattern, we installed

the PAA on a standing frame (Fig. 1b) for temporal use to radiate sound horizontally.
The measurements were made on fine (=no rain) days under calm wind (<2 m/s) with a
sound level meter set at a distance of 25 m, because a range of 10 m would be necessary
to complete producing audible sound from ultrasound with a PAA of this size (Prof.
Kamakura, 2018, personal communication). Safety was also considered for the
level-meter operators in determining the distance, as is discussed later. The PAA was
installed on top of a shed after the measurements (Fig. 1a). The audible sound pressure
level (SPL) pattern (Fig. 2) measured in the field indicated that the PAA exhibited high
directivity and low sidelobes, as expected; the SPL was less than 55 dB (dBA) at a
zenith angle of 40°, which was close to the value of the background noise level of 50 dB
despite the fact that the peak power (100 dB) was close to that of an acoustic speaker
(105 dB). By contrast, for the acoustic speaker, the SPL was as high as 70 dB even at a
zenith (elevation) angle of 85° (5°) and is therefore significantly more annoying to the
ear than a PAA.
To evaluate the parametric speaker for RASS measurements, temperature data
derived from the PAA-RASS were compared with values derived both from radiosonde
and from the acoustic speaker RASS. The dwell time for each RASS measurement was
set at about 57 s followed by an intermediate cessation operation time of 3 s, in which
the two speaker systems were alternately switched every minute for comparison. Each
RASS data set obtained with the two speaker systems was independently processed with
quality control to confirm the consistency in the height and time field values.
The profiles of virtual temperature derived from operational radiosonde
measurements were used as the standard reference data for comparison. The
radiosondes (the Meisei RS-11G used until September 2017, followed by the Meisei
iMS-100; Kizu et al., 2018) were launched from the Aerological Observatory, which is
located about 400 m northeast of the profiler (for the layout of the relative locations, see
Adachi et al., 2004). The time resolution of the radiosonde data used for the comparison
was 1 s, which corresponded to the height resolution of about 6 m. The radiosondes
were launched operationally at 08:30 JST (Japan Standard Time: JST=UTC+9 h), and
most of the RASS experiments included the launch time (Table 3). The RASS data were
taken during morning hours, on fine (= no rain) days, with light winds (< 3 m s$^{-1}$ at 20
m AGL), mostly in autumn, when the region was under the influence of a high-pressure
system. In the radiosonde comparison, the RASS data were averaged over about an hour
for each experiment to mitigate both the effects of vertical velocity (Angevine and
Ecklund, 1994; Görsdorf and Lehmann 2000) and the spatial difference between the
radiosonde and the profiler with RASS. Contrastingly, the 1 min raw RASS data were
used to compare the two speaker systems.

**3 Results of comparison**
**3.1 Applicability of parametric speaker to RASS**
As there are few, if any, studies on RASS using parametric speakers, preliminary
experiments were first conducted to confirm whether the secondary audible waves
produced by this type of speaker can propagate long distances along the radio wave
while satisfying the Bragg condition before evaluating it for RASS application. The
MRI PAA radiates bifrequency primary waves that are around 37 kHz and 40 kHz from
all the transducers simultaneously to generate the parametric sound of the secondary
difference frequency, which was around 3 kHz for RASS. Since sound absorption
generally increases with frequency, the ultrasound may be substantially dissipated as
altitude increases, although the peak SPL of the ultrasonic sound close to the PAA
(Table 2) was about 100 dB larger than that of audible sound generated by the acoustic
speaker (Fig. 2). The atmospheric absorption is a function of the sound frequency,
temperature, humidity, and pressure of the air (ISO, 1993). Example profiles of the
sound attenuation coefficient and attenuation at 3 kHz and 40 kHz derived from
radiosonde measurements are shown in Fig. 3. In the derivation, only the effect of
atmospheric absorption related to viscosity and thermal conductivity of the air,
molecular relaxation of rotation, and vibration of $O_2$ and $N_2$ was considered (see
Appendix), and other physical effects (e.g., reflection from the surface; ISO, 1996) were
disregarded. Figure 3a shows that the attenuation for the audible wave of 3 kHz
propagating from the surface to an altitude of 1 km above ground level (AGL) was 14.7
dB, which indicated that the sound wave at this frequency with an SPL of 105 dB on the
ground decreased to 90.3 dB at this altitude. By contrast, this figure also suggests that
the sound wave at 40 kHz with an SPL of 200 dB generated on the ground was reduced
to less than 0 dB at 160 m AGL. Thus, the primary wave of the PAA was not expected
to reach beyond this altitude. However, the difference-frequency component could
propagate to a higher altitude because it was audible sound.
Figure 4 shows a set of spectra obtained with the acoustic speakers and the PAA
at the time when the radiosonde measurement in Fig. 3 was made. The plots were
obtained by the LAP-XM, which is a software program developed on the basis of the
Profiler On-line Program (POP; Carter et al., 1995). The RASS echoes associated with
the acoustic speakers were obtained from altitudes as high as 1.3 km AGL. On the other
hand, those associated with the PAA were obtained from an altitude of 1.1 km AGL.
Although the PAA-RASS height coverage was somewhat lower than that associated
with acoustic speakers, this was much higher than the altitude where the primary
ultrasound waves were expected to dissipate. This result suggests that the secondary
difference-frequency component may reach the altitude comparable with the audible
wave generated by acoustic speakers while satisfying the Bragg condition and
propagating along the radio wave as an audible wave. The height coverage of the two
speaker systems are discussed later.

Another conformity of the secondary audible wave formed by the PAA to the

sound wave by the acoustic speaker for the RASS measurement can be seen in the
vertical profiles of the received echo power. Samples of the RASS echo power profiles
are shown in Fig. 5, along with profiles of radiosonde wind speed and horizontal
displacement of the sound beam center for RASS from that of the radio wave. The
samples were selected from the days (Table 3) when surface winds were light (< 2 m
$s^{-1}$) except on 19 October (Fig. 5a). The displacement of the sound wave with horizontal
wind was estimated by acoustic ray tracing based on radiosonde measurements. In this
estimation, the sound speed was estimated using Eq. (1), assuming a stationary
atmosphere, in which the virtual temperature was obtained from the radiosonde data,
and the initial displacement of the PAA from the profiler antenna on the ground was set
at 4 m (Fig. 1a). The RASS echo power shown here is a relative value, not absolute,
because the profiler is not calibrated for received power.

The RASS echo power of both speaker systems decreased with altitude except for

the first range gate. The reason for the decrease may include atmospheric attenuation of
the acoustic signal and displacement of the acoustic wave from the radar antenna by the
wind (Lataitis, 1992), as shown by the displacement profiles (Fig. 5). The echo power
with the acoustic speakers was almost always larger than that of the PAA (Figs. 5a–5d).
This could be explained by the acoustic speaker's larger peak power than that of the
PAA (Fig. 2), and the integrated peak power of the acoustic system, which comprises
four speaker units (Fig. 1), could be much larger. The echo power with the PAA was
slightly larger than that of the acoustic speakers at the first gate in Fig. 5a. This could be
because the sound from the PAA was advected above the antenna as shown by no
displacement at that height in the figure, suggesting that acoustic ray tracing was
reliable. The estimation of RASS echo power (e.g. Adachi et al., 1993) was beyond the
scope of this study. However, the echo power with both speaker systems in light-wind
conditions (Figs. 5b–5d) decreased almost linearly (in dB) with altitude above the first
gate, and the difference in the gradient between the two systems was relatively small
(less than 15 % on average), although this small difference may also be attributable to
the wind. From the facts mentioned above, we concluded that the secondary audible
waves formed by the PAA can propagate over a long distance along the radio wave
while satisfying the Bragg condition and are applicable to the RASS measurements as
the sound wave generated by the acoustic speaker.
Since the PAA was shown to be applicable to the RASS measurements, we next
explored the reliability of the PAA-RASS measurements by comparing with radiosonde
observations. It is noteworthy, however, that in Fig. 5a, the echo power with the PAA
decreased with altitude more sharply than that associated with the acoustic speaker at
altitudes between 300 and 700 m AGL, where relatively high winds were observed,
despite the fact that the PAA-RASS echo reached the highest range gate (1300 m AGL)
as the acoustic speaker RASS. This suggests that the PAA has enough peak power to
reach the highest range gate but is more susceptible to high winds than the acoustic
speakers. Thus, the effect of wind on the PAA-RASS measurements is discussed later in
this paper.

**3.2 Comparisons with radiosonde**
Profiles of virtual temperature ($Tv$) derived from radiosonde, the PAA-RASS, and the
acoustic speaker RASS observations are shown in Fig. 6 along with the corresponding
statistics for the data and the received power for both the PAA and acoustic speakers.
The RASS data were averaged over approximately an hour. The radiosonde data were
smoothed by 100 m running mean to match the RASS observations. The running mean
may also play a role in mitigating the effect of the temperature fluctuation due to
turbulence on the radiosonde measurements. The $Tv$ derived from the radiosondes was
in good agreement with the RASS measurements derived from both speaker systems,
lying within the error bar of most of the range gates. In addition, $Tv$ derived with both
speaker systems were close to each other. However, bias and standard deviation tended
to be large at inversion layers and at the first gate (e.g., Figs. 6a, 6b, and 6c), the latter
of which may corresponded to the smaller received power at that gate. This could be
attributable to the fact that the first gate is too close to the antenna. In fact, Lataitis
(1992) suggested that factors including the recovery of the receiver and incomplete
overlapping of the electromagnetic and acoustic beams due to the special separation
between the antenna and speaker systems can lead to a significant gradient in the
receiving power at this gate. In addition, a range error (e.g., Angevine and Ecklund,
1994; Görsdorf and Lehmann, 2000; Johnston et al., 2002) caused by the height
variation of the backscatter intensity may also contribute to the smaller received power.
It is also noteworthy that most of the highest range gates correspond to a received
RASS echo power of about $-10$ dB for both speaker systems in Figs. 5 and 6,
suggesting that the received power is one of the factors determining the height coverage,
although factors that determine the received power including the sound attenuation may
be different for each system.
Scatter diagrams comparing radiosonde virtual temperature with that from
RASS for all experiments are shown in Fig. 7 along with statistics. The first range gate
data of the RASS measurements were not considered because they are less reliable. This
figure shows that both the PAA and acoustic speaker RASS measurements of virtual
temperature were generally in good agreement with those derived from radiosonde
measurements, as expected. The linear regressions for both speaker systems were close
to the one-to-one relation, and correlation coefficients were close to unity. In addition,
the systematic error was less than 0.1 °C and the standard deviation was 0.4°C for both
systems, suggesting that both systems are reliable for RASS measurements.

**4 Discussions**
As reported above, we found many instances in which the PAA speaker system
exhibited comparable performance with the acoustic speakers with respect to the RASS
measurements in observing profiles of the Doppler spectrum and the virtual temperature,
as shown in the statistics for the comparisons both with radiosonde and with the
acoustic speaker RASS. Indeed, the bias and standard deviation for each speaker system
RASS with respect to radiosonde are in good agreement with results reported in
previous studies (e.g., Görsdorf and Lehmann 2000), despite no correction for vertical
velocity was done. This could be partly because the experiments were conducted on fine
days with light wind and because of the application of a relatively long averaging time.
In addition, removing the first gate data from the statistics may also have contributed to
the good results.

Although applying a long averaging time could mitigate the effect of vertical

airflow on bias (e.g., Moran and Strauch, 1994), it may degrade the statistics when the
virtual temperature profile evolves within the duration of the RASS measurement. On
the other hand, the statistics also indicated that the data number associated with the
PAA was smaller than that of the acoustic speakers (e.g., Fig. 6), implying that the
mean height coverage with the former was lower than that of the latter presumably
because of wind in addition to the low peak power mentioned previously (Fig. 2). Thus,
we independently focus our attention on both the effects of the time evolution of the
temperature profile on the statistics and of wind on the height coverage of the RASS
measurement in the following sections.

**4.1 Effect of rapid time evolution of temperature profile**
In the comparisons, the RASS data were averaged for a relatively long time to minimize
the effects of both vertical velocity and the spatial difference between the radiosonde
and the profiler with RASS. However, the temperature profiles derived from radiosonde
observations may not be well suited for use as standard reference data if the temperature
profile evolved rapidly within the hour-long RASS observation duration. In the
experiments, since the operational radiosondes were launched in the morning of fine
days with light winds, an inversion layer was frequently observed (Fig. 6). In fact, 12
inversions including multiple inversion layers (e.g., Fig. 6b) were observed in 8 of the
16 experiments. Inversion layers can evolve in a relatively short time due to surface
heating and cooling and/or the development of the boundary layer in the morning.
Indeed, the surface virtual temperature increased by 2.3°C on average with a standard
deviation of 1.0°C within an hour for the experiments shown in Fig. 6. Thus, the
temperature profile measured with the radiosonde can differ from the mean temperature
profile obtained from RASS even though both measurements represented an actual
profile, which may result in degrading the statistics for the RASS evaluation.

A sample of the temperature profile observing an inversion layer is shown in Fig.

8. This observation was made more than 3 h after sunrise (05:15 JST) on that day. The
$Tv$ profiles with error bars were the mean RASS measurements averaged over an hour
from acoustic (red) and PAA (blue) speakers. Both RASS profiles represented the
radiosonde profile to some extent but did not follow the profile well, especially around
the inversion layer. The large standard deviations indicated by long error bars may
reflect the time evolution of the temperature profile in addition to the measurement
precision of RASS. By contrast, the 1 min raw RASS data recorded around the
radiosonde launch time represented the inversion layer better than the mean RASS
measurements at some points, although there were still some discrepancies, which may
have been due to the locality of the inversion layer, the effects of vertical air motion or
turbulence, or the time difference between RASS and radiosonde in addition to the
accuracy and precision of the RASS measurements. The discrepancy above the
inversion layer may be caused by the locality of the temperature, because the MRI
observation field covered by vegetation (Adachi et al., 2005) ends about 500 m from the
profiler, which corresponds to the horizontal displacement at that height. On the other
hand, the discrepancies in and below the inversion could be mitigated by considering
the effect of the vertical airflow and/or applying a range correction. In terms of the time
difference, it is noteworthy that the radiosonde measurement is not a snapshot but
sequential; it took more than 2 min for the radiosonde to ascend to an altitude of 800 m
AGL, and the temperature profile may evolve even during this time. Thus, a comparison
with measurements that have both small spatial difference and high time resolution is
needed to evaluate the PAA-RASS measurement.

**4.2 Comparison with acoustic speaker RASS**

To suppress the effects of the spatial and time difference between the two

platforms on the evaluation, we next compared the temperatures derived from the
PAA-RASS with that from the acoustic speaker RASS. Of course, this comparison does
not provide an absolute but relative evaluation of the PAA-RASS measurement. This
issue should be kept in mind in examining the intercomparisons presented in this
section. In the intercomparison, the requirements for high-quality upper-air reference
data (bias $\leq$ 0.1K , $\sigma \leq$ 0.2K) proposed by WMO (2007) for the GRUAN were used as
criteria for the evaluation, although they are not for virtual temperature but for real
temperature.

A normalized frequency diagram and scatterplot of virtual temperature obtained

by the acoustic speaker RASS versus the PAA-RASS are shown in Fig. 9. The 1 min
raw data obtained alternately are presented in Fig. 9a, whereas the data averaged for
about an hour are plotted in Fig. 9b. Figure 9a shows that the PAA-RASS
measurements of virtual temperature were generally in good agreement with those of
the acoustic speaker RASS despite disregarding the time difference in the two systems.
The linear regression line was close to the one-to-one relation, and the correlation
coefficient was close to unity. Moreover, the mean bias and standard deviation of the
difference between the two speaker systems were less than 0.1°C and close to 0.4°C,
respectively, which are comparable with those obtained by the comparison with
radiosonde (Fig. 7) despite the higher time resolution. Since the spatial difference was
negligible and the time difference was quite small, the reason for this discrepancy could
include temperature fluctuation due to turbulence. Indeed, the mean (max and min)
increase of the virtual temperature at the surface for all the experiments was
0.2±0.5°C/10 min (1.4°C/10 min, −1.3°C/10 min), which suggests that temperature
fluctuation aloft was occurring.
A scatter diagram comparing the mean acoustic speaker RASS measurements
with those from the parametric speaker RASS is shown in Fig. 9b. The data were
averaged over about an hour to minimize the effect of temporal fluctuation of
temperature and improve the statistics. Indeed, the linear regression was close to the
one-to-one relation, and the correlation coefficient was closer to unity. In addition, both
the bias (0.06°C) and standard deviation (0.16°C) improved and satisfied the WMO
requirements.

From the evaluations mentioned above, we conclude that the accuracy and

precision of the parametric speaker RASS are comparable with those of the acoustic
speaker RASS for measuring the vertical profile of virtual temperature. The reliability
of the parametric speaker RASS could be improved by applying the time average over
the appropriate period, advanced quality control, and/or corrections for both range and
vertical airflow as long as the effect of the ground clutter is negligibly small.

**4.3 Effect of horizontal wind on the height coverage of the RASS measurement**
The reliability of the parametric speaker RASS measurement was shown to be
equivalent to the acoustic speaker RASS. However, we found many instances in which
the former tended to have less height coverage than the latter (Figs. 4, 5, and 6), which
is also reflected by the fewer number of data in the statistics (Figs. 6, 7, and 8).
Although the parametric speaker system exhibited less peak power than the acoustic
speaker system, the weak power cannot be the only reason for the lower height coverage
because the results show that the former can observe up to the highest range gate as the
latter as long as the received power is more than about -10 dB (e.g., Figs. 5a, 5b, 6a, and
8). On the other hand, the results also suggest that the reason may include the effect of
wind aloft (e.g., Fig. 5a). Because the acoustic beam generated by the parametric
speaker is narrow, it could be susceptible to the horizontal airflow, which displaces the
acoustic wave from the radar antenna as shown in Fig. 5. Thus, the effect of horizontal
wind on the height coverage of the parametric speaker RASS measurement was
evaluated by comparing it with the radiosonde wind data.

A scatter diagram comparing the mean RASS height coverage and horizontal

displacement of the center of the sound for RASS from that of radio wave at 1200 m
AGL is shown in Fig. 10, as well as the mean wind speed aloft. The horizontal
displacement was estimated by acoustic ray tracing. In the estimation, the initial
displacement of the acoustic speaker system from the profiler antenna on the ground
was set at 0 m, because the antenna is surrounded by the four acoustic speakers,
whereas that of the PAA was set at 4 m (Fig. 1a). The wind speed aloft is the mean
wind from 20 to 1200 m AGL (Table 3), which is the highest mean coverage of the
parametric speaker RASS measurements in calm wind conditions ($< 2$ m s$^{-1}$) as shown
in the figure. The data measured on 30 November 2016 are not considered in the
analysis because the RASS measurement was made more than 40 min later than the
radiosonde observation (Table 3). Note that the mean RASS height coverage shown in
the figure is different from the height coverage of the mean virtual temperature profile
in Fig. 6, because the latter reflects the maximum height coverage within the observed
profiles after quality control in the duration of the RASS measurement. The long error
bars may reflect the large time evolution of the RASS height coverage, which may also
be related to the evolution of the wind in the duration.

The parametric speaker RASS measurements tended to reach less altitude than the

acoustic speaker RASS, even when the horizontal displacement is less than 10 m
(corresponding to a wind speed of around 4 m s$^{-1}$). The reason for the lower coverage
under small displacement (light wind) conditions may include the parametric speaker's
lower peak power than that of the acoustic speaker system. The height coverage
decreased with the displacement and/or wind speed for the parametric speaker RASS, as
indicated by the linear regression analysis. In contrast, when the displacement is less
than 16 m (corresponding to a wind speed of around 6 m s$^{-1}$), most of the acoustic
speaker RASS measurements achieved a height coverage of around 1300 m AGL,
which was the highest range gate for the RASS measurement (Table 1). This suggests
that the acoustic speaker RASS keeps on observing at a high altitude even in relatively
high wind conditions, as also indicated by the short error bars.
It is noteworthy, however, that the height coverage of RASS with acoustic
speakers drops sharply to 1000 m AGL at a horizontal displacement of 15–16 m and
exhibits a tendency to decrease with the displacement afterward as the parametric
speaker RASS. By contrast, the height coverage of the parametric speaker tends to
decrease monotonically with the displacement at almost all ranges. These results
suggest that the parametric speaker RASS is more sensitive to wind because of the
narrow beam, whereas the acoustic speaker RASS is surprisingly robust. Since the four
acoustic speakers were not adjusted in phase, this robustness could be explained by the
higher aggregate sound power than that shown in Fig. 2 and possible location of sound
wave above the antenna in spite of relatively high winds.
To compensate for the lower wind tolerance, two additional experiments were
performed, in which the acoustic beam was broadened and steered. The parametric
speaker system employed for the RASS experiments was equipped with FPGA that
controlled the beam pattern of the sound, including beam width and direction. We
broadened the beam width from 5° to 12° (Fig. 2) when the parametric RASS echo was
observed up to an altitude of 1200 m AGL. However, this experiment resulted in a
decrease of the height coverage to 500 m AGL. The height coverage decrease could
have been due to the decrease of the peak power associated with beam broadening. In
fact, the measured peak power was decreased by 15 dB in our system by broadening the
beam (Fig. 2). Therefore, by using this technique, a parametric speaker with more peak
power was needed in our case to acquire equivalent height coverage with the acoustic
speaker system, which may result in increasing both the size and cost of the system.
On the other hand, the peak power does not decrease significantly with the zenith
angle of the beam as long as the angle is small. The SPL pattern at multiple zenith
angles measured in the field is shown in Fig. 11. The peak power was decreased by
about 7.5 dB by reducing the power supply to the PAA amplifier, which decreased not
only the audible sound but also the ultrasound levels for practical reasons (noisy) and
measurement safety. The results indicated that the peak power decreased by only 3.8 dB
when the beam was steered to a zenith angle of 10°, which corresponds to a horizontal
wind speed of 60 m s$^{-1}$. The sound wave might be displaced by the horizontal wind but
advected to above the antenna if the wave is generated windward with an appropriate
zenith angle. Thus, we conducted another experiment with the acoustic beam zenith
angle of 2° windward on a day when a mean wind speed of about 12 m s$^{-1}$ between 200
and 1200 m AGL was observed with the wind profiler. Unfortunately, no RASS echo
was observed, which may be partly because the sound wave did not propagate vertically
to the ground, and the advected sound wave front above the antenna was not normal to
the propagation direction of the radio wave. Additionally, the acoustic wave front may
have been distorted by wind shear. In that case, the radio wave might have been steered
to the direction normal to the sound wave front by considering the advection and
distortion of the sound wave front from the wind profiler measurements.

**4.4 Health effects of ultrasound exposure**
Since the ultrasonic SPL generated by the PAA is extremely high (>200 dB), the health
effects of ultrasound exposure in the area close to the PAA should be considered. In
studies involving small animals (WHO, 1982), mild biological changes have been
reported during prolonged exposure to airborne ultrasound with levels in the range of
95–130 dB at frequencies ranging 10–54 kHz, which become more severe with
increasing SPL. Thus, the PAA should not be installed on or under the ground level, as
it can be easily accessed by animals. Because the PAA for RASS emits sound vertically,
animals aloft, including birds and/or insects, can be exposed to the sound beam.
However, those animals are capable of avoiding the risk quite easily, because they can
perceive the audible sound from the PAA, and the beam width is very narrow. In fact,
no animals, including bugs and/or birds, died so far on the PAA after more than 100 h
of operation.
On the other hand, no adverse physiological or auditory effects appear to occur in
humans exposed to sound pressure levels up to about 120 dB (WHO, 1982; Health
Canada, 1991). At 140 dB, mild heating may be felt in the skin clefts. With increasing
sound pressure levels, the human body becomes warmer until death from hyperthermia.
This has been estimated to occur at levels greater than 180 dB. This lethal threshold
value corresponds to a distance of less than 17 m from the PAA, with an ultrasonic SPL
of 200 dB, assuming an atmospheric attenuation of 1.2 dB m$^{-1}$ (Fig. 3). To avoid
ultrasound exposure, we installed the PAA on top of a shed with a height of 2 m so that
the speaker won't be accessed by anyone. Moreover, rotational warning lights were
installed on the wall of the shed (Fig. 1d) to alert people to the emission of ultrasound
more than 50 dB (yellow) and/or 100 dB (red).

**5 Conclusions**
We investigated the applicability of parametric speakers to RASS for measuring
the vertical profile of virtual temperature by comparing the data with those obtained
from both radiosonde and the acoustic speaker RASS. In the experiments, the
operations of the two speaker systems were swapped every minute alternately for the
comparison. A detailed analysis of the profiles of both the acoustic attenuation and the
Doppler spectrum suggest that although the primary ultrasound generated by the
parametric speaker may be dissipated greatly as altitude increases, the secondary
audible waves generated from the bifrequency ultrasound can propagate long distances
while satisfying the Bragg condition.

We have also compared parametric speakers with both radiosonde and acoustic

speakers to estimate the reliability of RASS in measuring the virtual temperature ($Tv$).
The results indicated that $Tv$ measured with parametric speaker RASS has comparable
reliability with the acoustic speaker RASS measurements; the bias and standard
deviation (0.1°C, 0.4°C) for the parametric speaker were close to those for the acoustic
speaker (0.0°C, 0.4°C) with respect to radiosonde, which was consistent with the results
reported in previous studies, although the conditions in those studies, including the
corrections for the vertical wind and/or range, were different from ours. We also found
that not only the spatial difference between the two platforms but also both the
evolution of the temperature profile during the RASS measurement and temperature
fluctuation due to turbulence could contribute to deteriorate the statistics. To mitigate
these effects, a comparison of virtual temperature obtained from the two speaker
systems was also performed. The results indicated that the bias and standard deviation
(0.1°C, 0.2°C) of the parametric speaker RASS were quite small and satisfied the
requirements for high-quality upper-air reference data proposed by the WMO (2007).
Taken together, we conclude that parametric speaker RASS has comparable accuracy
and precision with acoustic speaker RASS with respect to the measurement of the
virtual temperature profile.

We examined the height coverage of RASS and found that the parametric speaker

deployed in the experiments tended to have less coverage than the acoustic speakers,
which may be a result of the parametric speaker having high directivity, and the
generated sound was more susceptible to the displacement from the radar antenna by
horizontal wind than the sound wave by the acoustic speakers. Thus, we broadened the
beam width of the parametric speaker, which resulted in degrading height coverage
because this operation deteriorates the peak power of the audible sound. The sound
wave was then steered windward with the default beam width (~5°) so that the advected
sound was located above the antenna. However, no echo was observed, presumably
because the sound wave front advected to above the antenna was not normal to the
propagation direction of the radio wave in the experiments. In addition, the sound wave
front may have been distorted by wind shear. This issue might be solved by using wind
profilers that are capable of steering the radio wave (e.g., Adachi and Kobayashi, 2001;
Law et al., 2002; Palmer et al., 2005) to the direction normal to the sound wave front as
Masuda (1988) proved with the MU radar (Fukao et al., 1985).
The results of this study including the statistics do not necessarily apply to all
locations, altitudes, and seasons; in particular, we note that the comparisons in this case
study were made in the morning on fine days with light wind when the effects of
horizontal and vertical wind would be less expected. Nevertheless, we confirm that a
parametric speaker is applicable to RASS measurement with a reliability comparable
with acoustic speakers. Although it is sensitive to horizontal wind, this type of speaker
could be installed to wind profilers located in urban areas for continuous-operational
observations (e.g., Ishihara et al., 2006) to improve weather forecast because it has high
directivity and no horizontal sound wave leaks to annoy nearby residents.

*Acknowledgements.*
The authors wish to thank Emeritus Professor T. Tsuda of Kyoto University for many
helpful discussions and comments regarding the research presented and Mr. S. Hoshino
of the Aerological Observatory for providing information on radiosonde measurements.
The first author wishes to thank Mr. J. Neuschaefer of Vaisala for offering the LAP-XM
software to analyze the spectrum data, Prof. T. Kamakura of the university of
electro-communications, Mr. S. Onogi of the meteorological instrument centre, Mr. N.
Okushima and Mr. E. Suzuki of Starlite Co., Ltd. and Mr. T. Takai for technical support,
and Drs. Y. Shoji, M. Mikami, A. Segami, S. Tsunomura, and Mr. T. Sakashita of MRI
for providing an opportunity to conduct the experiments. The authors also thank
anonymous reviewers who made many helpful comments that improved this work
substantially. This study was partially supported by JSPS KAKENHI Grant Number
15K01273 and 17H00852.

**Appendix**
**Calculation of the atmospheric attenuation**
The method of estimating attenuation coefficient for atmospheric absorption from
temperature, humidity, and pressure is summarized here based on ISO 9613-1 (ISO,
1993). The attenuation coefficient $\alpha$ (dB m$^{-1}$) is expressed by the sum of four terms in
good approximation as
$$\alpha = \alpha_{cl} + \alpha_{rot} + \alpha_{vib,O} + \alpha_{vib,N},$$   (A1)
where $\alpha_{cl}$ represents the classical absorption caused by the transport processes, $\alpha_{rot}$ is
the molecular absorption by rotational relaxation, and $\alpha_{vib,O}$ and $\alpha_{vib,N}$ indicate the
molecular absorption caused by vibrational relaxation of oxygen and nitrogen,
respectively. The molecular absorption by other compositions of the air including
carbon dioxide is small and neglected in the calculation.

The first two terms of Eq. (A1) related to the classical and rotational absorption is

given by their sum, $\alpha_{cr}$
$$\alpha_{cr} = \alpha_{cl} + \alpha_{rot} = 1.60 \times 10^{-10} \left(\frac{T}{T_0}\right)^{\frac{1}{2}} \left(\frac{P_a}{P_r}\right)^{-1} f^2,$$   (A2)
where $T$ (K) is the atmospheric temperature, $T_0$ is the reference air temperature (293.15
K), $P_a$ (hPa) is the atmospheric pressure, $P_r$ (hPa) is the reference air pressure (1013.25
hPa), and $f$ (Hz) is the sound frequency.

The two vibrational relaxation terms in Eq. (A1) are given respectively by,

$$\alpha_{vib,O} = [(\alpha\lambda)_{max,O}] \times \frac{f}{c_s} \times \left\{ 2\left(\frac{f}{f_{rO}}\right) \left[1 + \left(\frac{f}{f_{rO}}\right)^2\right]^{-1} \right\},$$   (A3)
and
$$\alpha_{vib,N} = [(\alpha\lambda)_{max,N}] \times \frac{f}{c_s} \times \left\{ 2\left(\frac{f}{f_{rN}}\right)\left[1 + \left(\frac{f}{f_{rN}}\right)^2\right]^{-1}\right\} ,$$ (A4)
where subscripts O and N represent oxygen and nitrogen, respectively, $[(\alpha\lambda)_{max}]$ (dB
m$^{-1}$) represents the maximum attenuation by a vibrational relaxation over a distance of a
wavelength, $\lambda$ (m), $c_s$ (m s$^{-1}$) is the sound speed, and $f_r$ (Hz) is the relaxation frequency.

The maximum attenuation by a vibrational relaxation for oxygen and nitrogen are

given respectively by,
$$[(\alpha\lambda)_{max,O}] = \left(\frac{40\pi}{35}\right)(log_{10}e)X_O\left(\frac{\theta_O}{T}\right)^2 exp\left(-\frac{\theta_O}{T}\right),$$ (A5)

$$[(\alpha\lambda)_{max,N}] = \left(\frac{40\pi}{35}\right)(log_{10}e)X_N\left(\frac{\theta_N}{T}\right)^2 exp\left(-\frac{\theta_N}{T}\right),$$ (A6)
where $X_O$ (= 0.209476) and $X_N$ (=0.78084) represent the standard molar concentrations
of dry air, and $\theta_O$ (=2239.1 K) and $\theta_N$ (=3352.0 K) are the characteristic vibrational
temperature for oxygen and nitrogen, respectively.

The sound speed $c_s$ in Eq. (A3) and (A4) at a molecular concentration of water

vapor of $h$ (%) is given by
$$c_s = c_a \times \sqrt{1 - \frac{h}{100}\left(\frac{\gamma_w}{\gamma_a} - \varepsilon\right)} = c_0 \times \sqrt{\frac{T}{T_0}} \times \sqrt{1 - \frac{h}{100}\left(\frac{\gamma_w}{\gamma_a} - \varepsilon\right)} ,$$ (A7)
where $C_a$ is the sound speed for dry air, $\gamma_w$ (=1.33) and $\gamma_a$ (=1.40) are heat capacity ratio
for water vapor and dry air, respectively, $\varepsilon$ (=0.662) is the ratio of the molecular weight
of water vapor to the molecular weight of air, and $C_0$ is the sound speed for dry air at
the reference air temperature, $T_0$. The value of $h$ is given from the relative humidity, $h_r$
(%) by
$h = h_r \left(\frac{P_{sat}}{P_r}\right) / \left(\frac{P_a}{P_r}\right) = h_r \left(\frac{P_{sat}}{P_a}\right)$ ,                    (A8)
where $P_{sat}$ (hPa) is the saturation vapor pressure given by
$P_{sat} = P_r \times 10^{\left(-6.8346 \times \left(\frac{T_{01}}{T}\right)^{1.261} + 4.6151\right)}$ ,             (A9)
and $T_{01}$ (=273.16 K) is the triple-point isotherm temperature. The sound speed in dry air
$C_a$ is given by
$c_a = \sqrt{\frac{\gamma_a R}{M_d} T}$ ,                       (A10)
where $R$ (= 8.314 J mol$^{-1}$ K$^{-1}$) is the universal gas constant, and $M_d$ (=2.896×10$^{-2}$ kg
mol$^{-1}$) is the molecular weight for dry air. By substituting values of $R$, and $M_d$ to Eq.
(A10), we may derive
$c_a = 20.048\sqrt{T}$ ,                  (A11)
and $C_0 = 343.25$ m s$^{-1}$ at a temperature of $T_0$. Note that Eq. (A11) corresponds to Eq. (1)
in stationary atmosphere because air temperature $T$ is equal to virtual temperature $T_v$ in
dry air.

The relaxation frequency for O and N are given by,

$f_{rO} = \left(\frac{P_a}{P_r}\right)\left(24 + 4.04{\times}10^4 h \frac{0.02+h}{0.331+h}\right)$ ,            (A12)
and
$f_{rN} = \left(\frac{P_a}{P_r}\right)\left(\frac{T}{T_0}\right)^{-\frac{1}{2}} \times \left[9 + 280h + exp\left[-4170\left\{\left(\frac{T}{T_0}\right)^{-\frac{1}{3}} - 1\right\}\right]\right]$ ,            (A13)
respectively.

By substituting Eqs. (A2) — (A11) to Eq. (A1), we may derive,

$\alpha \approx 8.686f^2 \left[\left\{1.84{\times}10^{-11}\left(\frac{P_a}{P_r}\right)^{-1}\left(\frac{T}{T_0}\right)^{\frac{1}{2}}\right\} + \left(\frac{T}{T_0}\right)^{-\frac{5}{2}} \times \left\{0.01275{\times}exp\left(\frac{-2239.1}{T}\right)\right\}\left\{f_{rO} + \right.\right.$
$\left.\left(\frac{f^2}{f_{rO}}\right)\right\}^{-1} + 0.1068{\times}exp\left(\frac{-3352.0}{T}\right)\left\{f_{rN} + \left(\frac{f^2}{f_{rN}}\right)\right\}^{-1}\right]$, (A14)
where $f_{rO}$ and $f_{rN}$ are given by Eqs. (A12) and (A13), respectively.

The attenuation coefficients at 3 kHz and 40 kHz as a function of temperature and

relative humidity estimated using Eq. (A14), is shown in Fig. A1. This figure indicates
that the attenuation coefficient for ultrasound at 40 kHz is larger than that for audible
sound at 3 kHz, as expected. In addition, the attenuation coefficient depends on the
temperature and humidity for both frequencies. Note that the attenuation coefficient for
audible sound peaks at lower temperatures (<10°C) than those for ultrasound,
suggesting that the attenuation coefficient could increase with altitude for the former,
while it decreases for the latter (e.g., Fig. 3, >1 km AGL, where $T = 20.2°C$ and $h_r =$
76% near the surface). In contrast, the contribution of air pressure to the attenuation
coefficient on the ground does not differ very much from that at an altitude of 1100 m
AGL (~900 hPa).

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

**List of Tables**

**List of Figures**

frequency of 3 kHz. The error bars represent 2σ. The SPL pattern for the acoustic
speaker in the negative zenith angle region is a mirror image of the pattern measured at
the positive zenith angle for ease of viewing. The background noise level was about 50
dB. The SPL was measured with a sound level meter (Rion NL-42).

Figure 3. Profiles of atmospheric-attenuation coefficient $\alpha$ and atmospheric attenuation
for sound at frequencies of (a) 3 kHz and (b) 40 kHz derived from the radiosonde
measurements at 08:30 JST on 19 October 2016, at the MRI site.

Figure 4. Doppler spectra from RASS observations measured with (a) acoustic speakers
from 08:30 JST for 1 min and (b) the parametric speaker from 08:31 JST for 1 min on
19 October 2016. At each height, the first moment of the spectrum, indicated by the
vertical bar, gives the vertical sound velocity, and the second moment, indicated by the
horizontal bar, gives the spectral width.

Figure 5. Profiles of received mean RASS echo power, horizontal displacement of the
parametric speaker sound from radio wave, and wind speed on (a) 19 October 2016, (b)
27 October 2016, (c) 9 August 2017, and (d) 7 September 2017, derived with the
acoustic speakers (red), the parametric speaker (blue), and radiosonde (green). The error
bars represent $2\sigma$. The black lines indicate linear regressions for the received power data
(except for the first range) as shown in the upper-right legend with correlation
coefficients.

Figure 6. Profiles of virtual temperature ($Tv$) and received power ($Pr$) from 08:30 JST
on (a) 15 October, (b) 21 October, (c) 8 November, and (d) 30 November 2016 derived
from a radiosonde (black), RASS with acoustic speakers (red and orange), and the
parametric speaker (blue). The radiosonde data were smoothed by 100 m running means
to match with the vertical resolution of the RASS. The error bars represent $2\sigma$ in the
RASS hourly observations. The mean, standard deviation, and number of samples of
temperature difference are summarized in a table in each panel.

Figure 7. Scatterplots of virtual temperature of the RASS vs. the radiosonde
measurements at all heights except for the first range. The data derived from the RASS
with the acoustic speakers (the parametric speaker) are plotted as open (closed) circles.
The radiosonde data were smoothed by 100 m running means to match with the vertical
resolution of RASS. The lines represent linear regressions for each data set as shown in
an upper legend along with the correlation coefficients. The mean, standard deviation
and number of samples of temperature difference are summarized in a bottom table.

Figure 8. Profiles of the virtual temperature ($Tv$) from 08:30 JST on 7 September 2017,
derived from a radiosonde (black), RASS with acoustic speakers (red), with the
parameter speaker (blue), and horizontal displacement of the radiosonde from the
profiler (green). The radiosonde data were smoothed by 100 m running means to match
with the vertical resolution of RASS. The error bars represent 2σ in the RASS
observations averaged over 60 min, and closed circles represent 1 min raw data from the
time indicated. The mean, standard deviation, and number of samples of temperature
difference of RASS from radiosonde are summarized in the table.

Figure 9. Comparisons of the parametric speaker vs. the acoustic speakers in measuring
virtual temperature at all heights (except for the first gate) shown by (a) a normalized
frequency diagram (color scale) and (b) a scatterplot. The data obtained from each
speaker system every 1 min alternately were used in (a), whereas the hourly-mean data
were plotted in (b). The mean *Tv* derived with the acoustic speakers is shifted 10°C for
ease of viewing in (b). The lines represent linear regressions for each data set, shown in
the upper-left and lower-right legends along with correlation coefficients, respectively.
The mean, standard deviation, and number of samples of temperature difference are
summarized in each table.

Figure 10. Scatterplots of mean height coverage of RASS measurement vs. horizontal
displacement of the beam center of the sound for RASS from that of the radio wave at
1200 m AGL derived from radiosonde observations. Closed circles (squares) denote the
observed mean RASS height coverage by acoustic speakers (parametric speaker) with
standard deviations indicated by error bars. The color scale represents the mean wind
speed aloft (20–1200 m AGL). Thick lines represent linear regressions for each data set,
where the PAA data are divided by a height threshold of 1100 m AGL. The highest
range gate sampled for the RASS measurement is 1300 m AGL.

Figure 11. Audible sound pressure pattern of the parametric speaker at a frequency of 3
kHz measured at multiple zenith angles, shown in the upper legend with the beam width
observed. Note that the peak SPL was decreased by about 7.5 dB for safety. The SPL
was measured with a sound level meter (Rion NL-42).

Figure A1. Simulated atmospheric-attenuation coefficients for sound at the frequencies
of (a) 3 kHz and (b) 40 kHz as a function of the atmospheric temperature and the
relative humidity at an atmospheric pressure of 1013 hPa. Results for a pressure of 900
hPa are also plotted for a relative humidity of 20%.

Table 1. Parameters of the wind profiler with RASS.

| | |
|---|---|
| Frequency | 1357.5 MHz |
| Peak Power | 500 W |
| Beam width | < 7° |
| Beam elevation | 90° |
| Pulse width | 665 ns |
| First range gate | 200 m |
| Last range gate | 1300 m |
| Gate spacing | 100 m |
| Interpulse period | 12163 ns |
| Coherent Integration | 3 |
| Spectra Averaged | 191 |
| Number of FFT points | 8192 |
| Acoustic source | Pseudo-random frequencies (random hop) |
| Location | 36°03'19"N, 140°07'28"E |
| Manufacture | Scintec |
| Model | LAP-3000 |

Table 2. Characteristics of the MRI parametric speaker.

| | |
|---|---|
| Center frequency | 40.0±1.0 kHz |
| Band width (-6dB) | < 2.0 kHz |
| Sound pressure level | > 200 dB (at 0.3m, 40 kHz, theoretical value) |
| Number of transducers | 10008 |
| Number of channels | 278 |
| Beam width | 5°—17° (1° step) |
| Beam elevation | 60—90° (1° step) |
| Beam azimuth | 0—359° (1° step) |
| Input audio signal freq. | 2.8—3.3 kHz |
| Speaker diameter | 1.8 m |
| Speaker system size | 2.1 x 2.1 x 1.8 m |
| Manufacture | Starlite Co., Ltd. |
| Model | 100FM-001 |

Table 3. List of the comparison experiments, including date, period, sea level pressure, surface temperature, surface wind speed, and mean wind speed aloft (20 — 1200 m AGL) with standard deviation. Means and standard deviations are not vector but scalar statistics.

| Date | Time [JST] | $P_{sea}$ [hPa] | T [°C] | U [m s$^{-1}$] | $\overline{U}_{aloft}$ [m s$^{-1}$] |
|---|---|---|---|---|---|
| 14 Oct. 2016 | 0803—0900 | 1023.0 | 15.0 | 2.0 | 5.8±2.8 |
| 15 Oct. 2016 | 0801—0900 | 1025.5 | 15.8 | 1.4 | 2.1±0.5 |
| 19 Oct. 2016 | 0801—0900 | 1015.6 | 20.2 | 3.1 | 5.4±1.7 |
| 21 Oct. 2016 | 0801—0900 | 1016.5 | 15.7 | 2.1 | 3.1±0.7 |
| 24 Oct. 2016 | 0819—0900 | 1014.9 | 13.1 | 2.0 | 3.2±1.5 |
| 27 Oct. 2016 | 0837—0932 | 1016.1 | 18.8 | 0.8 | 1.6±0.6 |
| 28 Oct. 2016 | 0803—0900 | 1018.4 | 12.5 | 1.2 | 7.6±3.3 |
| 31 Oct. 2016 | 0809—0900 | 1024.3 | 12.7 | 1.7 | 5.4±2.6 |
| 02 Nov. 2016 | 0825—0902 | 1023.2 | 9.8 | 1.6 | 5.0±2.3 |
| 08 Nov. 2016 | 0803—0902 | 1017.0 | 7.4 | 2.2 | 5.4±3.4 |
| 12 Nov. 2016 | 0809—0906 | 1019.8 | 11.8 | 0.4 | 3.7±1.3 |
| 30 Nov. 2016 | 0911—0932 | 1030.4 | 5.7 | 1.8 | 2.9±1.4 |
| 29 Mar. 2017 | 0843—0900 | 1020.0 | 8.4 | 2.4 | 4.9±1.3 |
| 09 Aug. 2017 | 0845—0900 | 991.3 | 30.5 | 1.2 | 4.5±2.5 |
| 07 Sep. 2017 | 0801—0900 | 1003.2 | 22.3 | 1.8 | 2.3±0.5 |
| 09 Apr. 2018 | 0825—0902 | 1013.6 | 10.2 | 1.6 | 6.9±5.1 |

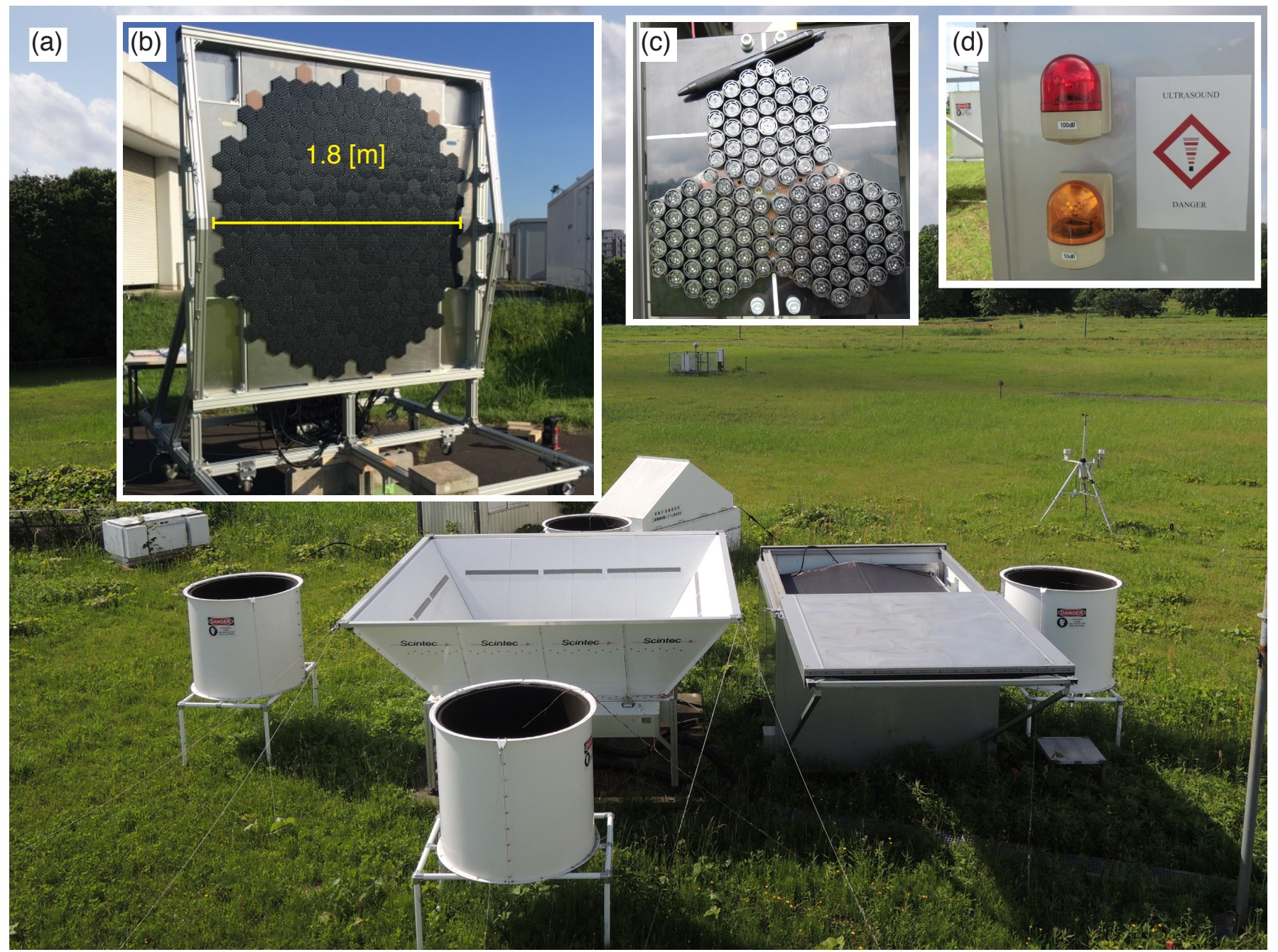

Figure 1. Pictures of (a) LAP-3000 with acoustic speakers and a parametric speaker for RASS, (b) overview on a support frame, (c) partial expanded view of the parametric speaker, and (d) rotary warning lights on the shed wall. The parametric speaker mounted on top of the shed with a sliding roof is covered with rainproof film in the field, as shown in (a).

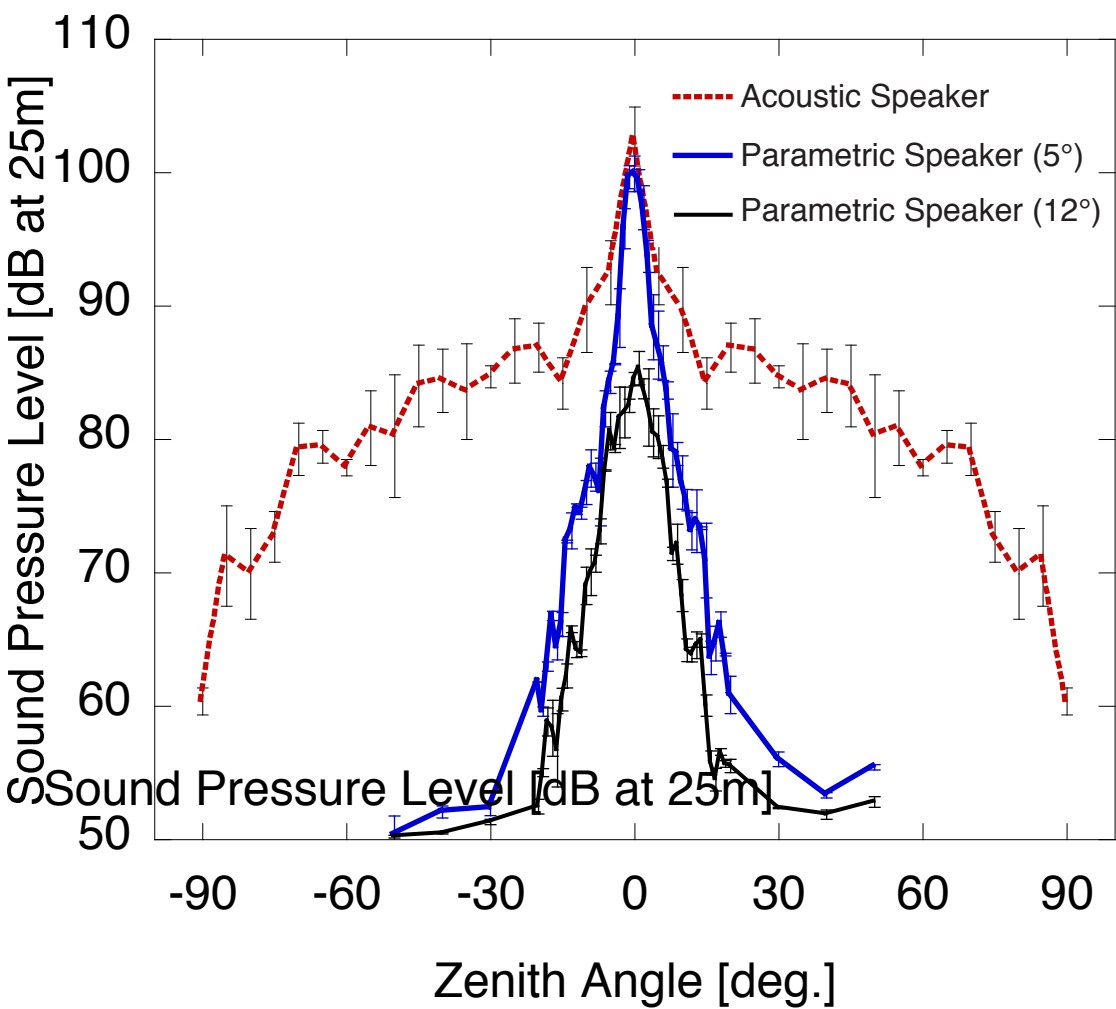

Figure 2. Audible sound pressure level (SPL) pattern for an acoustic speaker (red), the parametric speaker with the measured beam width of 5° (blue) and 12° (black) at a frequency of 3 kHz. The error bars represent 2σ. The SPL pattern for the acoustic speaker in the negative zenith angle region is a mirror image of the pattern measured at the positive zenith angle for ease of viewing. The background noise level was about 50 dB. The SPL was measured with a sound level meter (Rion NL-42).

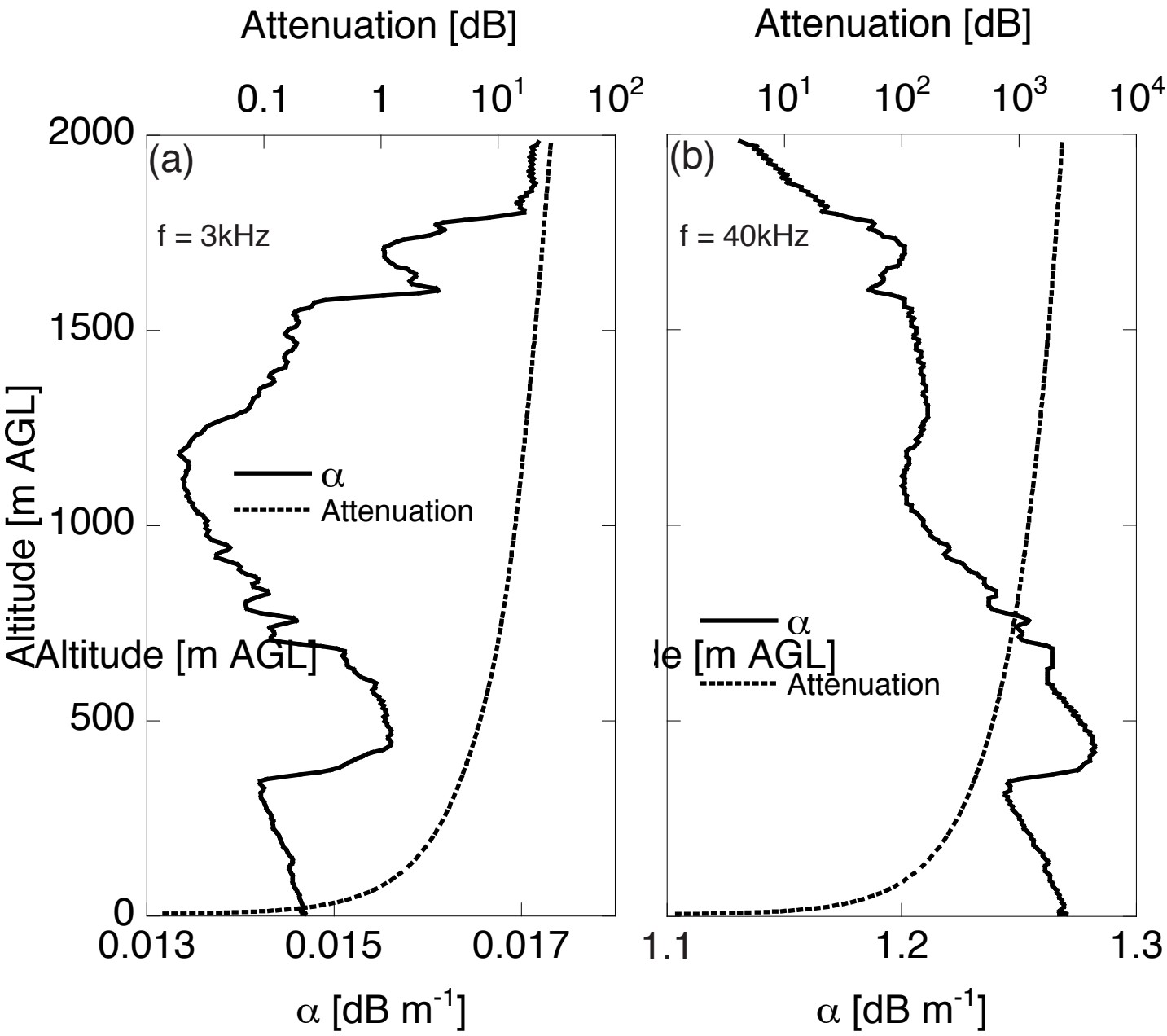

Figure 3. Profiles of atmospheric-attenuation coefficient α and atmospheric attenuation for sound at frequencies of (a) 3 kHz and (b) 40 kHz derived from the radiosonde measurements at 08:30 JST on 19 October 2016, at the MRI site.

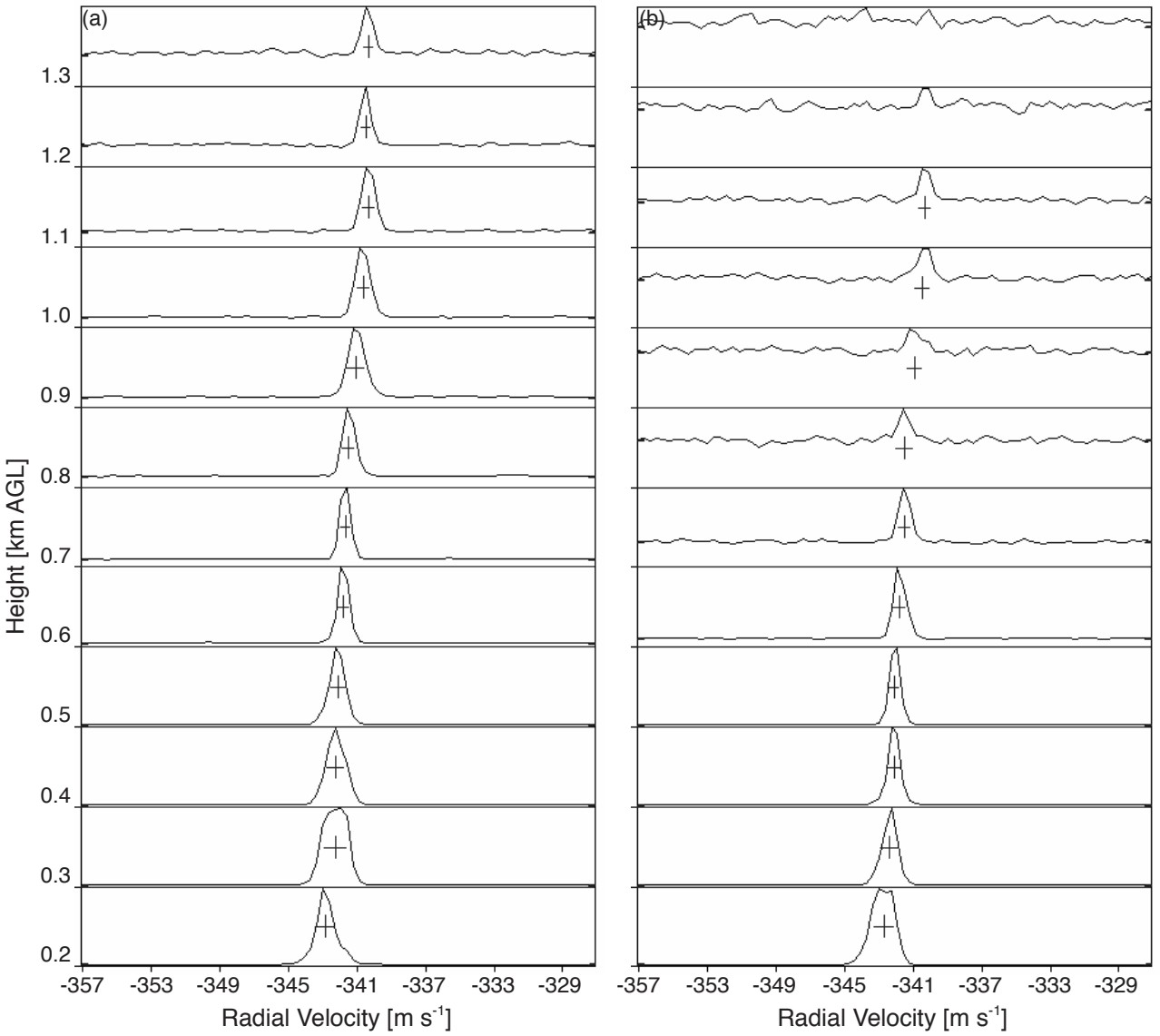

Figure 4. Doppler spectra from RASS observations measured with (a) acoustic speakers from 08:30 JST for 1 min and (b) the parametric speaker from 08:31 JST for 1 min on 19 October 2016. At each height, the first moment of the spectrum, indicated by the vertical bar, gives the vertical sound velocity, and the second moment, indicated by the horizontal bar, gives the spectral width.

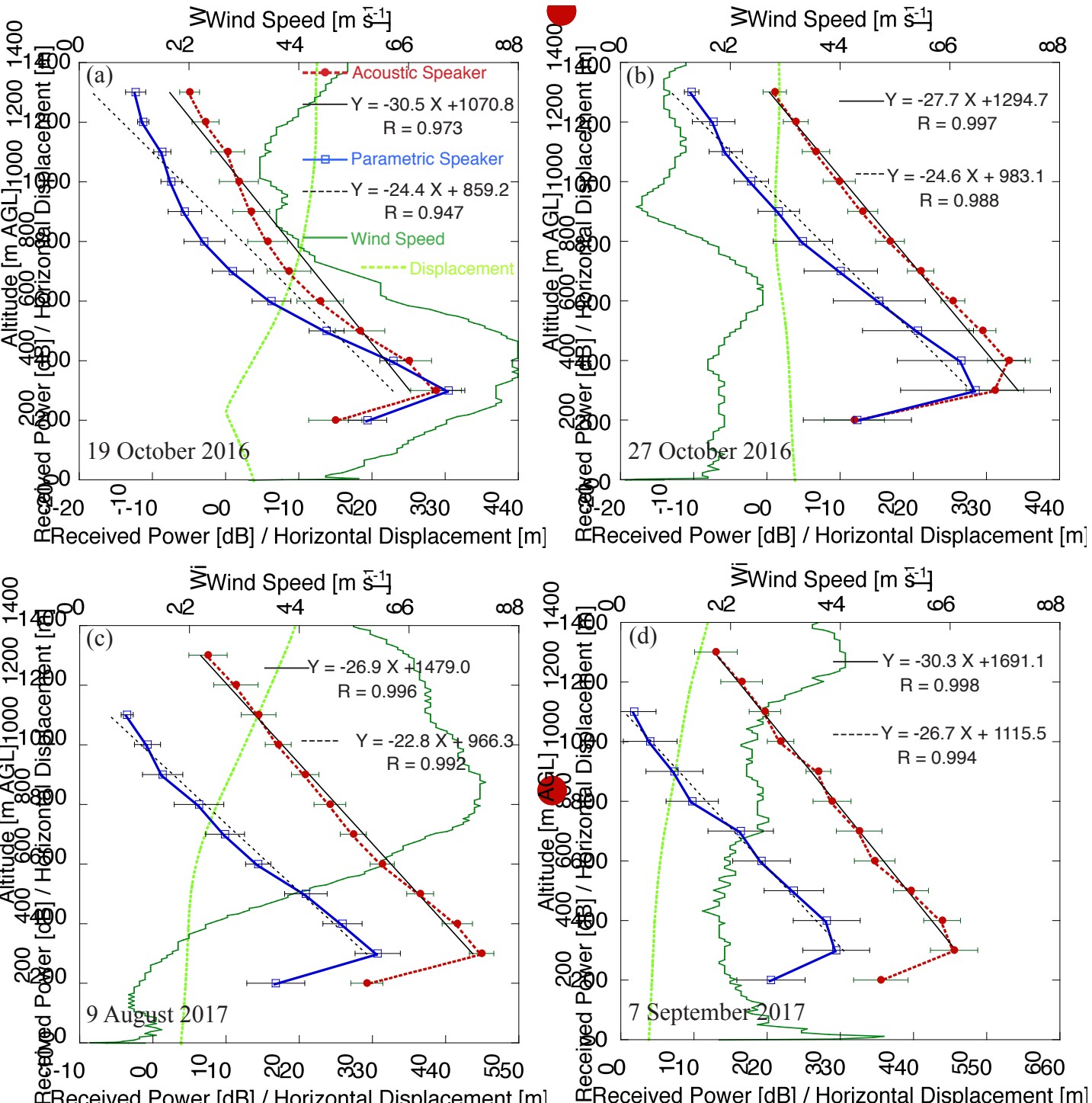

Figure 5. Profiles of received mean RASS echo power, horizontal displacement of the parametric speaker sound from radio wave, and wind speed on (a) 19 October 2016, (b) 27 October 2016, (c) 9 August 2017, and (d) 7 September 2017, derived with the acoustic speakers (red), the parametric speaker (blue), and radiosonde (green). The error bars represent 2σ. The black lines indicate linear regressions for the received power data (except for the first range) as shown in the upper-right legend with correlation coefficients.

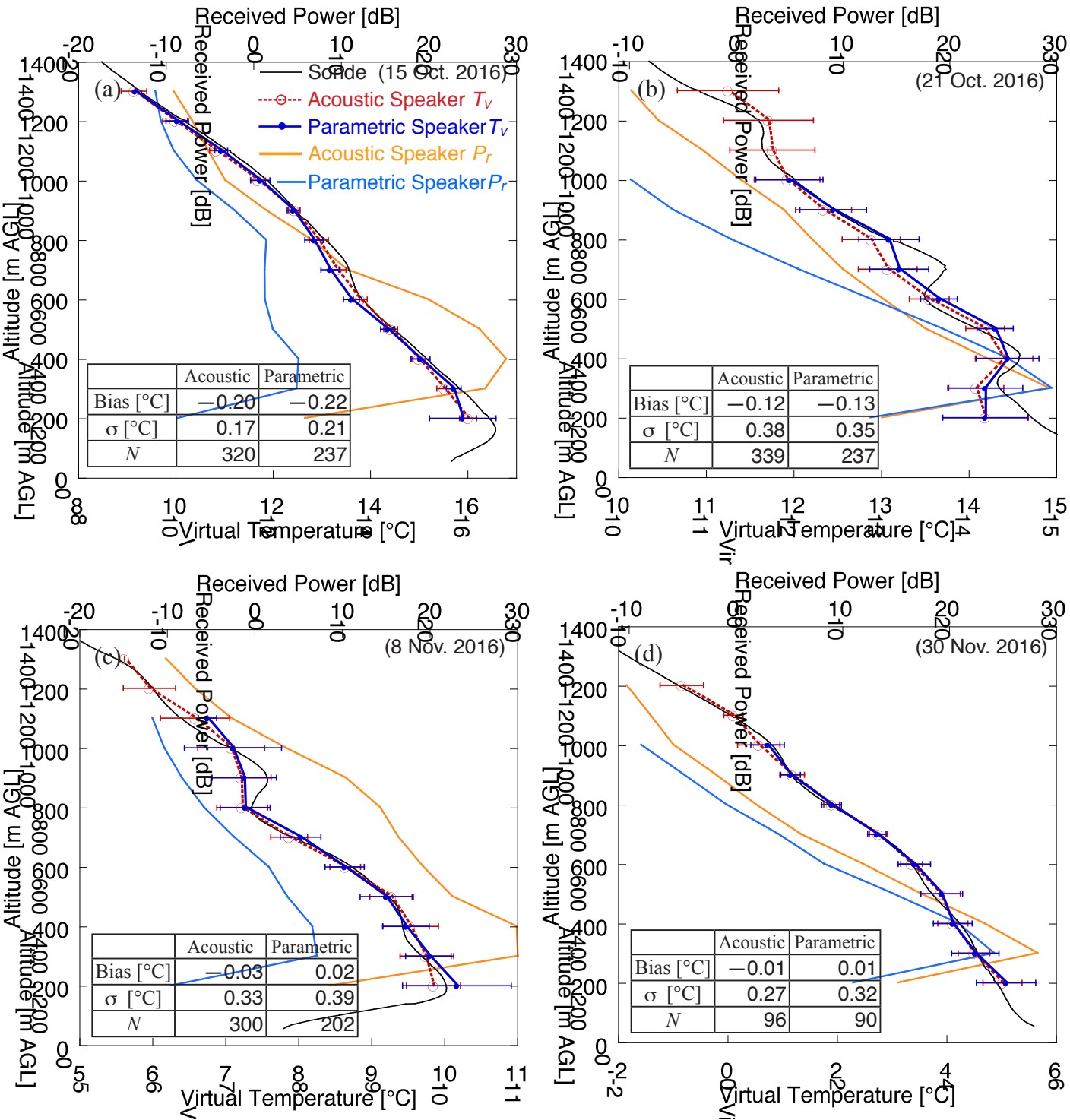

Figure 6. Profiles of virtual temperature ($T_v$) and received power ($P_r$) from 08:30 JST on (a) 15 October, (b) 21 October, (c) 8 November, and (d) 30 November 2016 derived from a radiosonde (black), RASS with acoustic speakers (red and orange), and the parametric speaker (blue). The radiosonde data were smoothed by 100 m running means to match with the vertical resolution of the RASS. The error bars represent 2σ in the RASS hourly observations. The mean, standard deviation, and number of samples of temperature difference are summarized in a table in each panel.

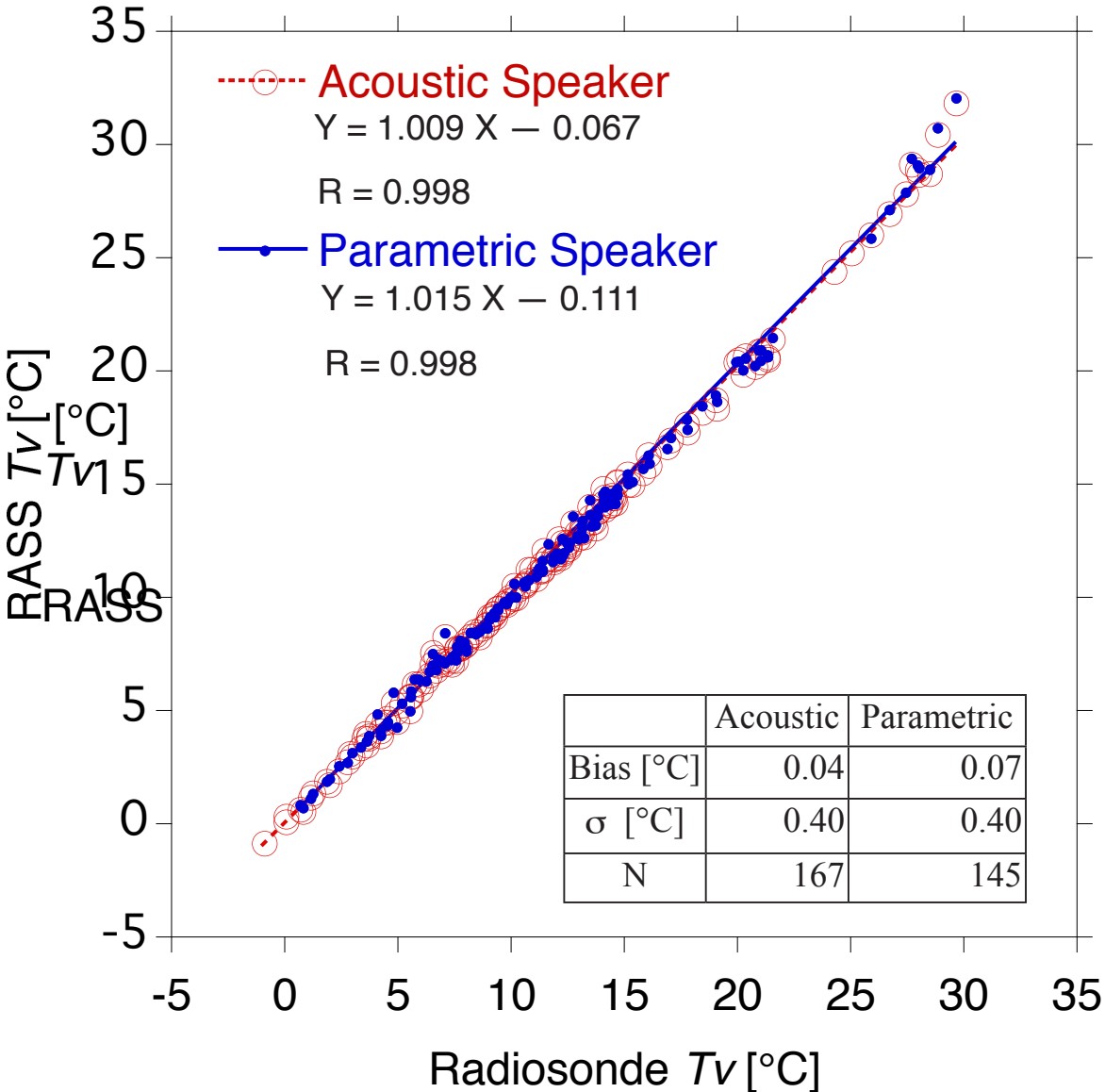

Figure 7. Scatterplots of virtual temperature of the RASS vs. the radiosonde measurements at all heights except for the first range. The data derived from the RASS with the acoustic speakers (the parametric speaker) are plotted as open (closed) circles. The radiosonde data were smoothed by 100 m running means to match with the vertical resolution of RASS. The lines represent linear regressions for each data set as shown in an upper legend along with the correlation coefficients. The mean, standard deviation and number of samples of temperature difference are summarized in a bottom table.

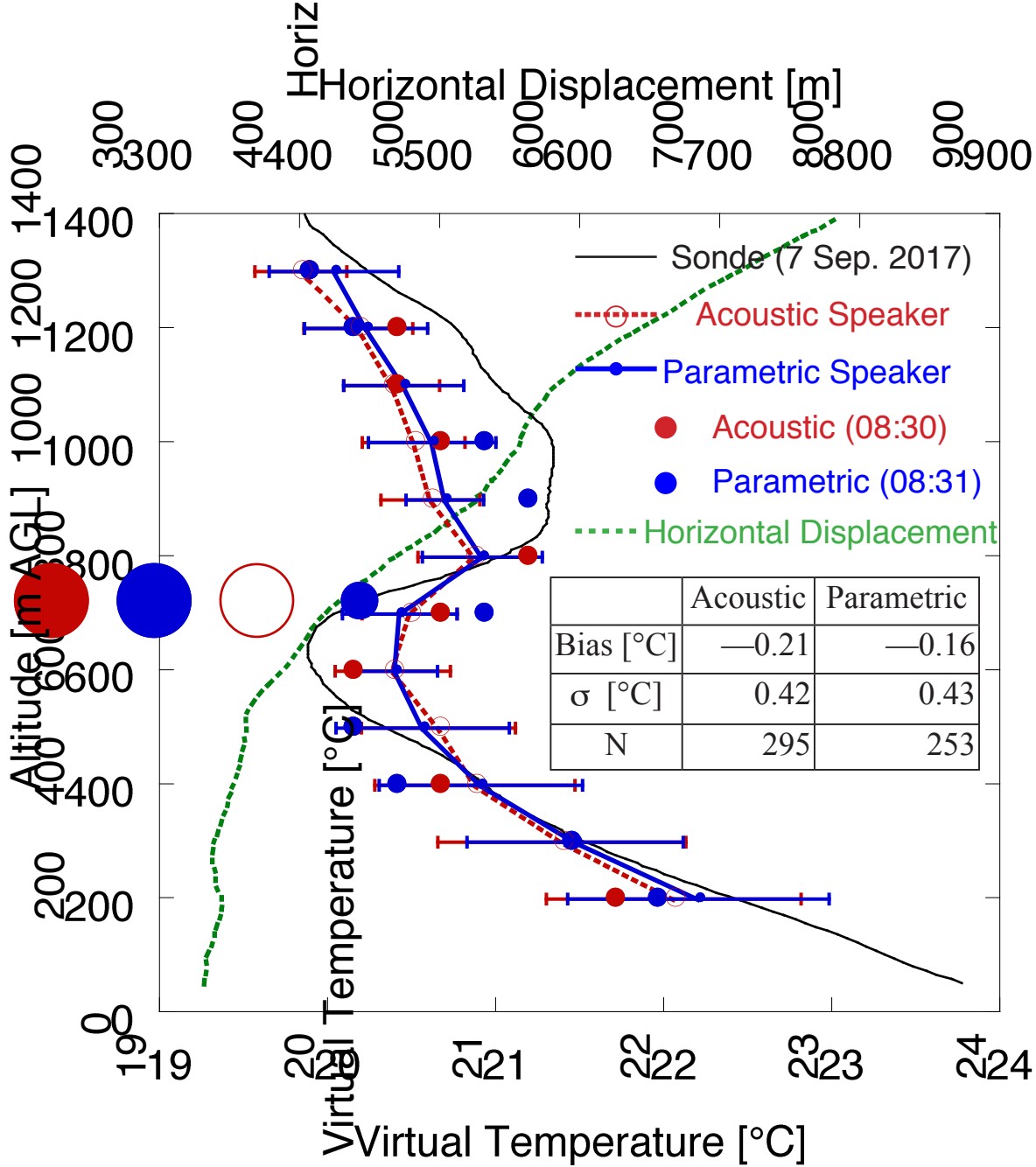

Figure 8. Profiles of the virtual temperature ($T_v$) from 08:30 JST on 7 September 2017, derived from a radiosonde (black), RASS with acoustic speakers (red), with the parameter speaker (blue), and horizontal displacement of the radiosonde from the profiler (green). The radiosonde data were smoothed by 100 m running means to match with the vertical resolution of RASS. The error bars represent 2σ in the RASS observations averaged over 60 min, and closed circles represent 1 min raw data from the time indicated. The mean, standard deviation, and number of samples of temperature difference of RASS from radiosonde are summarized in the table.

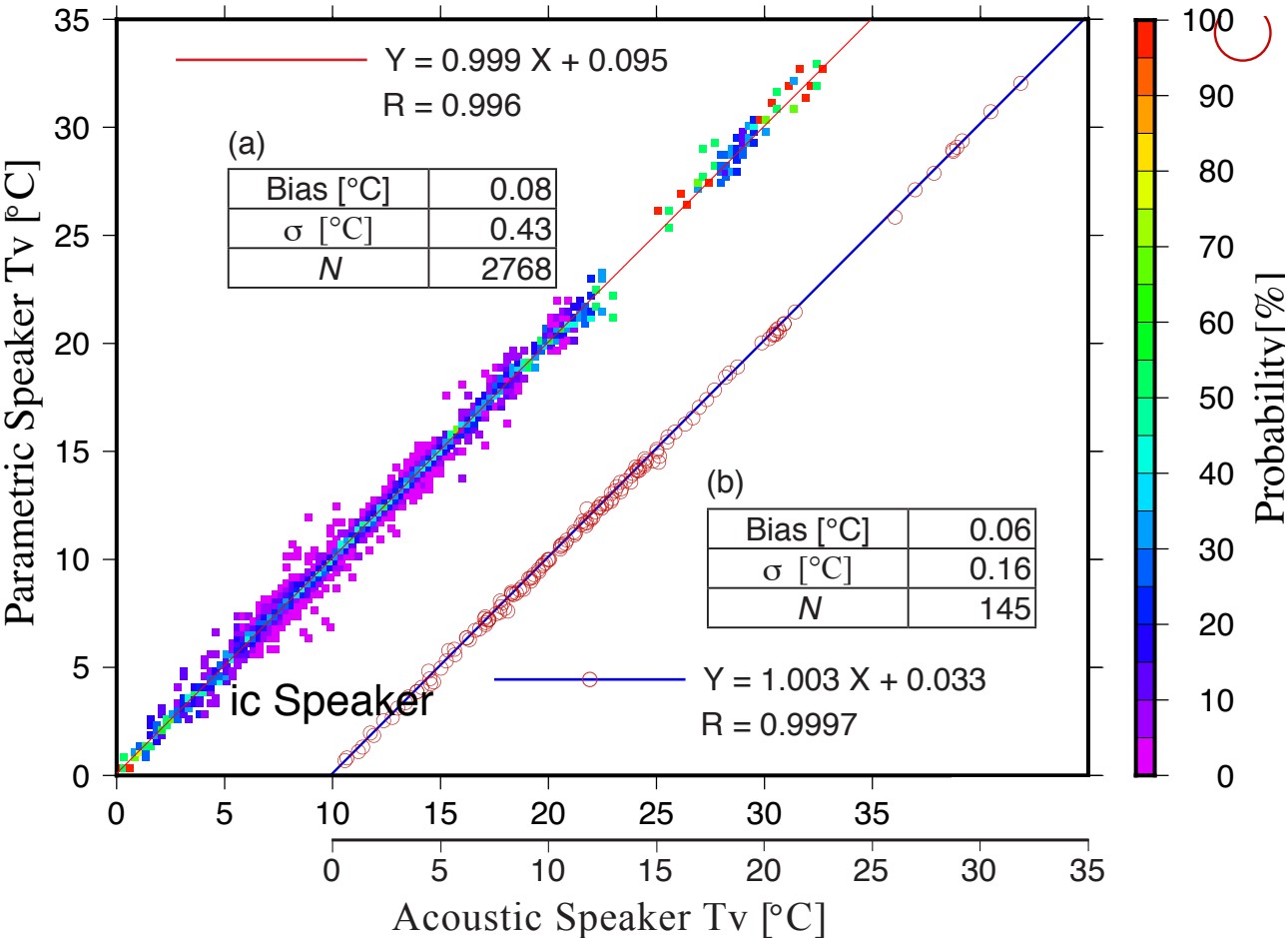

Figure 9. Comparisons of the parametric speaker vs. the acoustic speakers in measuring virtual temperature at all heights (except for the first gate) shown by (a) a normalized frequency diagram (color scale) and (b) a scatterplot. The data obtained from each speaker system every 1 min alternately were used in (a), whereas the hourly-mean data were plotted in (b). The mean Tv derived with the acoustic speakers is shifted 10°C for ease of viewing in (b). The lines represent linear regressions for each data set, shown in the upper-left and lower-right legends along with correlation coefficients, respectively. The mean, standard deviation, and number of samples of temperature difference are summarized in each table.

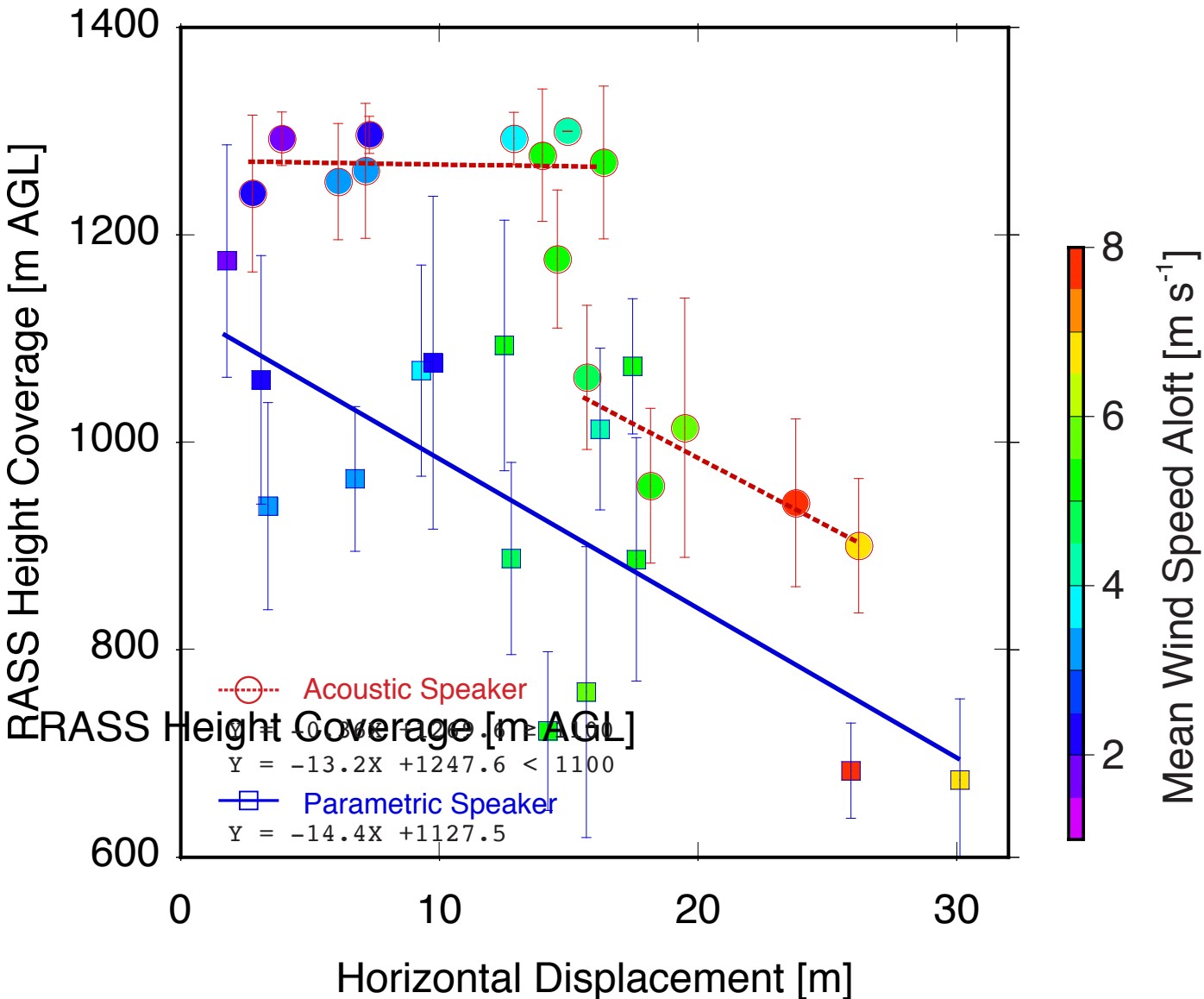

Figure 10. Scatterplots of mean height coverage of RASS measurement vs. horizontal displacement of the beam center of the sound for RASS from that of the radio wave at 1200 m AGL derived from radiosonde observations. Closed circles (squares) denote the observed mean RASS height coverage by acoustic speakers (parametric speaker) with standard deviations indicated by error bars. The color scale represents the mean wind speed aloft (20–1200 m AGL). Thick lines represent linear regressions for each data set, where the PAA data are divided by a height threshold of 1100 m AGL. The highest range gate sampled for the RASS measurement is 1300 m AGL.

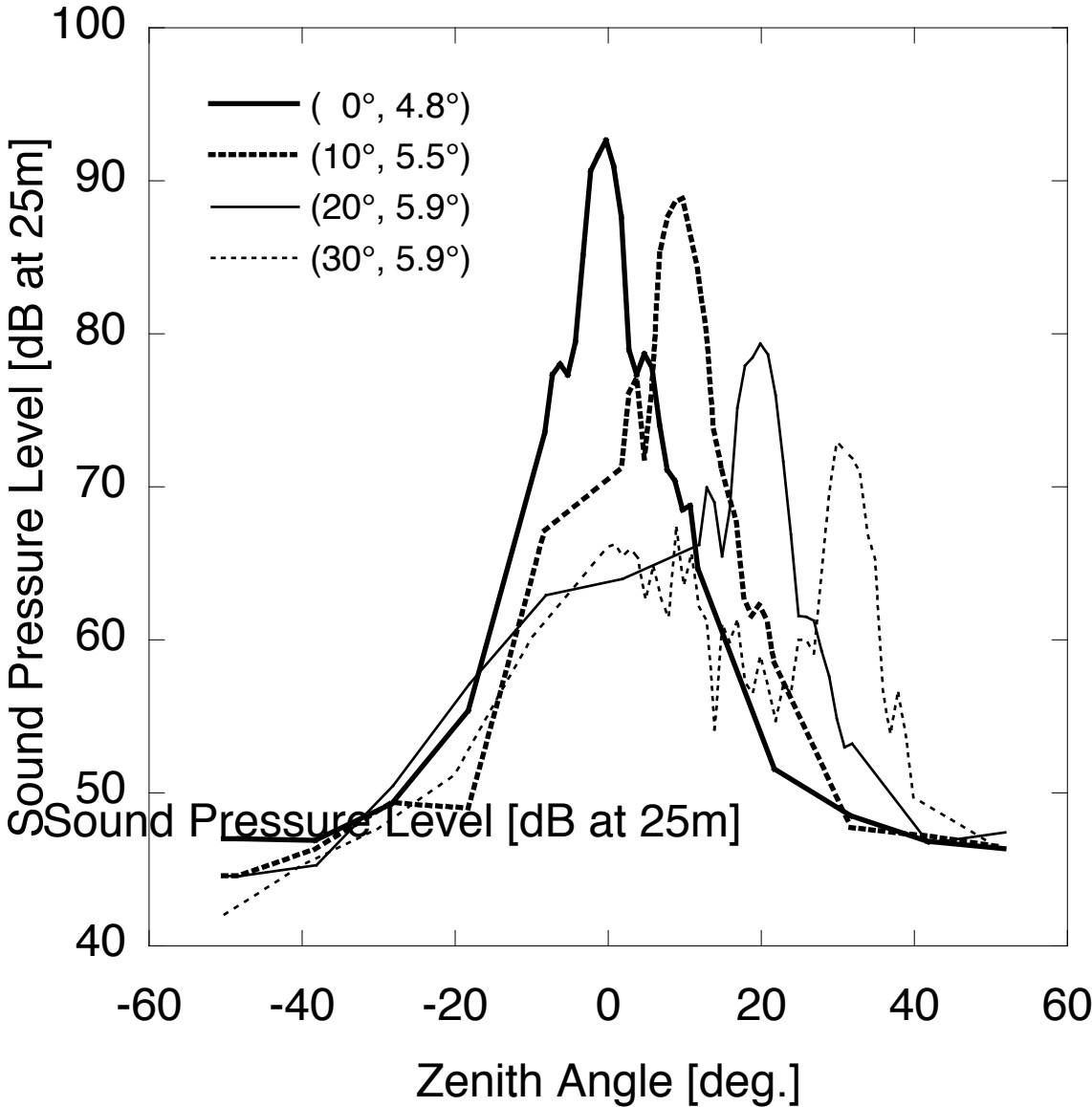

Figure 11. Audible sound pressure pattern of the parametric speaker at a frequency of 3 kHz measured at multiple zenith angles, shown in the upper legend with the beam width observed. Note that the peak SPL was decreased by about 7.5 dB for safety. The SPL was measured with a sound level meter (Rion NL-42).

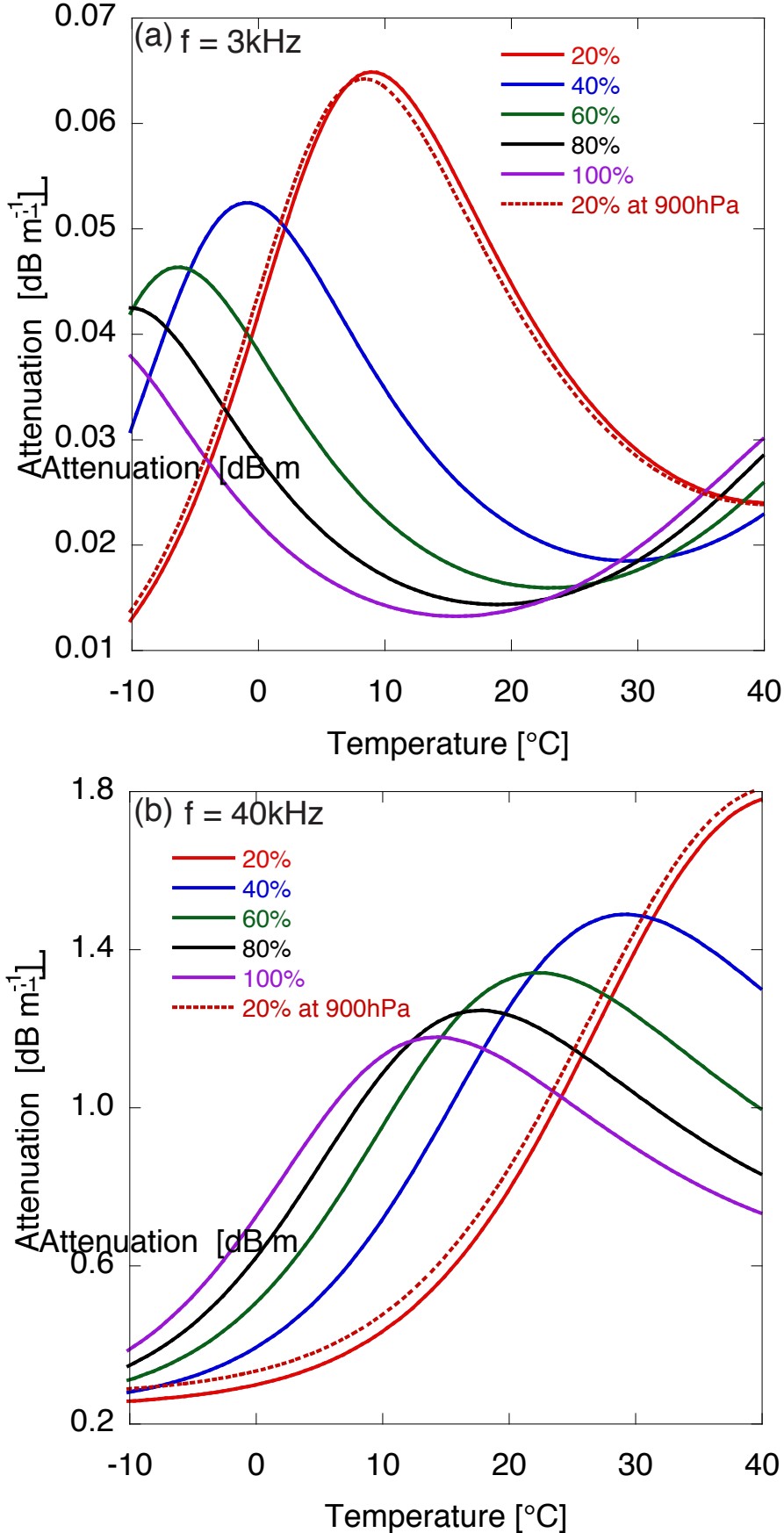

Figure A1. Simulated atmospheric-attenuation coefficients for sound at the frequencies of (a) 3 kHz and (b) 40 kHz as a function of the atmospheric temperature and the relative humidity at an atmospheric pressure of 1013 hPa. Results for a pressure of 900 hPa are also plotted for a relative humidity of 20%.

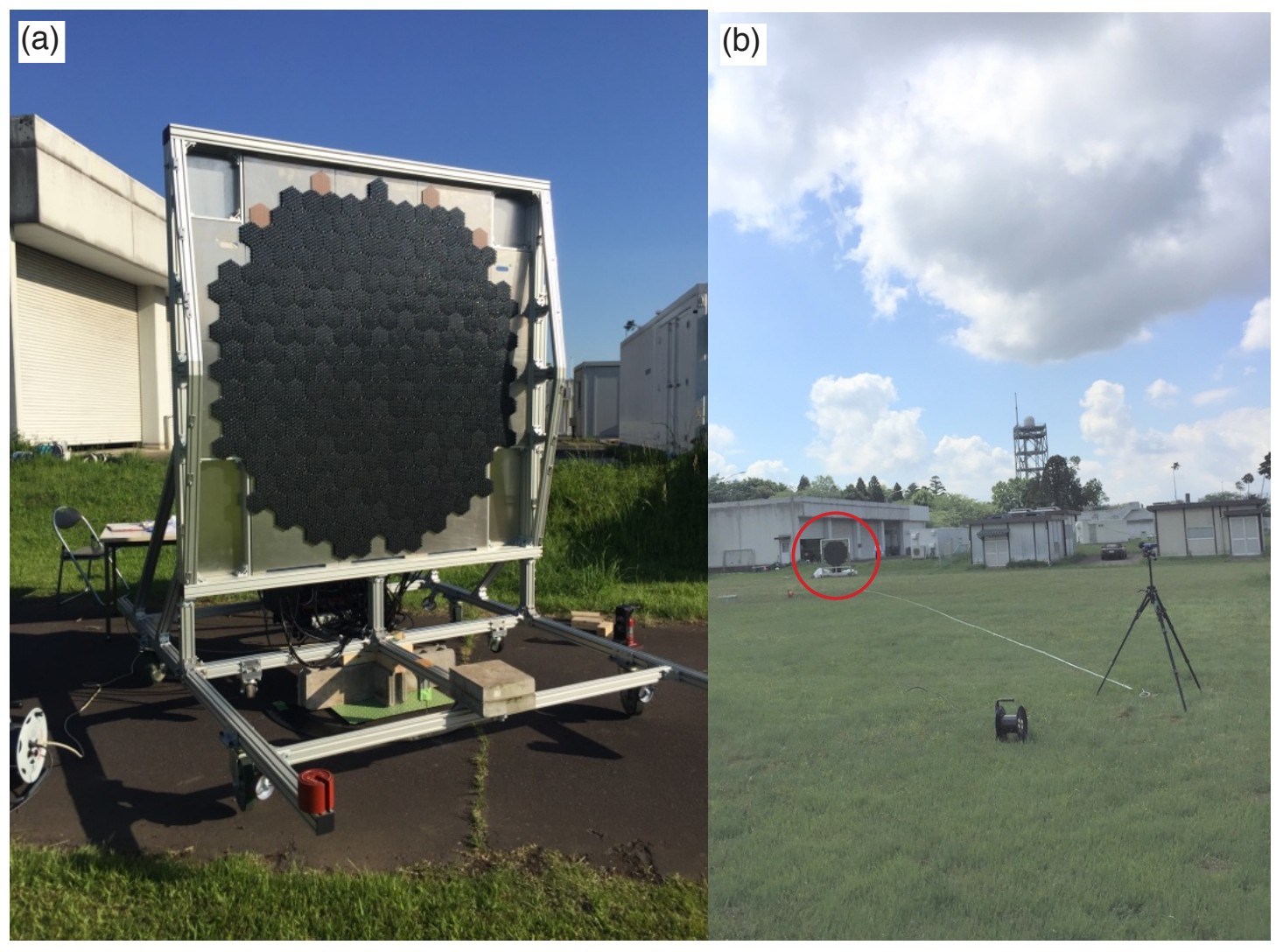

Fig. 2R. The PAA (a) on a support frame and (b) in the field to measure the audible sound pressure level pattern. A circle in red indicates the location of the PAA 25 m apart from a sound level meter on a tripod in (b).