# Peer review of "Application of Parametric Speakers to"

_Atmospheric Measurement Techniques, 2019_

## Referee Comment (RC1) · Anonymous Referee #1 · 10 May 2019

Review of : **'Application of parametric speakers to radio acoustic sounding system' from A. Adachi and H. Hashiguchi, in AMTD.**

**Overall recommendation**: Minor revisions required

**General comments:**

The paper presents a new method to use the RASS system for wind profilers. This novel system gets around one major defect of the RASS which cannot be used continuously in inhabited areas due to the noise pollution. The parametric speakers allow to measure the secondary difference frequency of ultrasonic waves emitted at 37 and 40 kHz. Ultrasonic waves are rapidly absorbed in the air, whereas the secondary wave of around 3 kHz are transported and backscattered like the wave of classical RASS. The characteristics of the parametric speakers are their low side lobes (an advantage) and their higher directivity (perhaps a disadvantage, I'm not sure to be able to conclude). The paper describes an accurate comparison between both systems, in order to assess the results of the parametric system.

The manuscript fits the standard of AMTD and is well organized. The abstract reflects the content. The problem is clearly introduced although some points have to be improved. The paper is well documented although I find that the works from other authors are sometimes too quickly cited and would deserve to be developed further. The instrumentation is well described and the comparison is accurately and seriously conducted. Figures are useful, clear and accurately described.

I've got a unique reservation. I did not experience RASS systems myself but I heard about the difficulty to retrieve the level of inversion layers when individual profiles of RASS are compared to radiosonde profiles. These difficulties are hidden by time averages and uncertainties calculation on the whole profiles. The authors mention these difficulties, cite authors who explain how to cope with them, but I think it is not enough to make the reader trust the method. I make some proposals in the following to clarify this point. Note that I do not question the results that are presented in this manuscript.

**Specific comments, minor remarks, suggestions:**
**Page 2**, line 24: I do not understand the use of 'if' in this sentence.
Line 29: I would write 'versus' instead of '/' in 'RASS/radiosonde'.

**Page 3**, line 47: 'measures' → I suggest 'provides'
Line 48: I would add 'from the measurement of the Doppler shift between the emitted and backscattered radar wave', after 'propagates vertically'
Line 49: 'among' → 'between'

**Page 4**, lines (54-55): This issue about the vertical velocity correction is very important and not well presented. It is obvious that w has to be included. The fact that you could not use w is an issue, and you show in the following how you cope with it (1-hour averages). The way the issue is introduced, at the beginning of the paper, is not satisfying especially the sentence : 'the vertical wind speed is considered in Eq (1) because the neglect of the wind velocity ….'. I explain why.
At the beginning (p 4), it is obvious that the correction has to be done. You mention p 7 (lines 112-115), that w could not be used due to the automobiles echo. So I deduced you had used the oblique beams instead of the vertical one. I understood in the following that you did not.
Please, could you clarify this issue in your presentation, perhaps by saying at the beginning that you had to neglect w and that this issue would have to be discussed?

If you cite May's and Angevine's papers it could be interesting to add (further on in the manuscript) some details of their discussion, including the discussion on the downward bias in the profiler velocity measurements mentioned in Angevine et al., 1998: the 1-hour average is a way to mitigate this effect.

Minor comment: in lines 57, 59 and 61 respectively, you use brackets for two different purposes: with 'bias (systematic error)', 'standard deviation (precision)', the brackets provide the meanings whereas in 'bias (standard deviation)', you present an alternative. The latter use of the brackets is continued through the manuscript. The first use could be presented in a different way.

Line 63 : I would add 'accurate' before 'corrections'
Line 64: if relevant, I would add 'thermodynamic' before 'constants'

Lines 61 to 64: I was not convinced by the results you cite (Görsdorf's and Lehman's work) as long as I had not read the paper. My reservation was about the one year time average that reduced the bias as expected. When reading this paper, I was convinced and I found additional interesting information that would deserve to be discussed in your own paper (but probably not in the introduction):
- the reason for the range correction would be helpful to comment (see my point, Page 16, line 245). Görsdorf and Lehman are not the only ones who mention this range correction: Angevine et al., 1998 also do, but in the references you cite, Görsdorf and Lehman discuss the benefit of the correction at the inversion levels, which is a major advance and could feed your discussion.
- the distinction between climatological investigation and individual profile comparison is also an important point to comment: you can hardly compare errors amounts when they are computed with varying time-scales (unless you discuss the conditions).

To conclude, I would say that the use of this reference comes too early and that you could take more advantage in discussing it further on.

In general, I suggest that you give more details about the results and the uncertainties that are reported in the literature, concerning the RASS assessment. For instance, Angevine et al., 1998 made a comparison of Tv during 1 month, at one level (396m agl) between a RASS system (UHF) and tower instruments. The uncertainties that they reported are expected to be different from those obtained for instance by Moran and Strauch (1994) who used a RASS with a VHF profiler, during 5 weeks (400 profiles, compared to RS). Moreover, in each case, some information should be given about the corrections (vertical velocity correction, range correction …).

**Page 5**, line 66:  you could also add 'the distortion of the acoustic wave due to turbulence and vertical temperature gradients (Lataitis, 1992)'

Line 68 : 'A wind profiler/RASS has been used' → Joined measurements of wind profiler and RASS have been frequently used'.

Line 71: 'Important limitations of this method include the emission' → 'Among the limitations of this method, an important one is the emission'

Line 80 : 'is expected'  → 'would be promising'?

**Page 7**, line 100: please provide the latitude of the site
Line 106: 'The MRI wind profiler' → 'The MRI 1357-MHz wind profiler'
Line 113: see my previous remarks about w (p. 4)

**Page 8**, line 119: 'In the experiments' → 'For the experiment purpose'
Line 122-123: about pseudorandom frequencies. This is either too much or not enough explained for a non-expert reader. Is it necessary to mention? I suggest to move this specificity to Table 1.

**Page 9**, line 134-137: am I right to say that the beamwidth of the speaker should be larger than the MRI beamwidth? Does that mean that you start with degraded conditions with the default sound beam? Is it the point you raised when you spoke of the parametric speaker directivity (line 22, page 2)? I raise this point since I wondered at the beginning whether the directivity would be an advantage or a drawback.

Line 137: turbulence will broaden the sound beam width, especially inside the boundary layer, but it will also reduce the measurement range, as you said before.

Line 137: 'aloft' is not accurate enough and I wonder whether it is appropriate here.

Line 138: 'measured in the field' I would add: 'with a sound level meter'. Could you describe the measurement protocol?

Line 143: 'and is **therefore** significantly more annoying' (therefore can be added if you find it relevant).

**Page 11**, line 161: at 10-m height?

Lines 163-167 : my first reaction is that the launching occurred in a period of the day when the time temperature gradient is important and the boundary layer top level is rapidly increasing. You discuss this point later on, but you could already add a few comments. I also suggest to add a column in Table 3, with the time difference between the launching and the sunrise time. I calculated 2h30 to 4h. Am I right?

Line 170: I do not understand the meaning of 'availability'. Do you mean 'practical relevance'?

**Page 12**, line 182: tell me if I am wrong, but it seems to me that the reference (ISO 1993) is not easily available. Could you provide (in an appendix) the equations to calculate the sound attenuation coefficient and the attenuation, derived from the RS measurements?

**Page 13**, line 200: 'the PAA also reached a minimum altitude of 1.1 km AGL' → 'the PAA reached an altitude of 1.1 km AGL'.

Lines 203-206: this unique example is not enough to draw a conclusion. So, I would use 'may reach' instead of 'can reach' and indicate that the propagation level will be systematically compared between the 2 systems.

or, you can simply say that the example in Fig. 4 is promising….

**Fig. 5**:  usually, the backscattered signal of wind profiler radars is artificially corrected at the first gate to avoid receiver saturation. Are you sure it is not the case for RASS systems? Could this be a reason for the increase of the echo power between the first and the second gate?

**Page 15**, line 228: 'available' → 'convenient'

Line 231: 'immediately' → 'sharply'

**Fig. 6**: I suppose your sample of profiles is the best you could provide. Would it be possible to have a look on the corresponding Received Power profiles?

Why is there such a large difference in the number of data (320 and 237 in Fig. 6a) while the measurement range is the same for both systems? On the opposite, in Fig. 6d, 96-90=6 is small while the PAA vertical range is 200 m lower than the acoustic speaker. I probably miss something.

I suppose you tried comparisons with RS data averaged during 15 min or 30 min. Were the results so bad?

**Page 16**, line 245: '(e.g. Figs 6n and 6c)' → 6a also.
Line 249 : I suggest that you give here the reason for the range corrections presented by Görsdorf and Lehmann and that you briefly explain the processing.
The range corrections should be applied to the whole profile (and not only at the first gate), to mitigate the discrepancies at the inversion levels.

Lines 251 to 253 (page 16, 'The RASS data were averaged for about an hour. The RS data were smoothed by 100 m running means to match the RASS observations)' should be moved p15, just before the comments on Fig. 6.

**Page 18**, line 278: as said before for Fig. 6, I'm not convinced by the argument of the link between the smaller data number for the PAA and the height coverage. In addition, the wind is relatively stronger at 1300m in Fig. 5a, while the height coverage is similar for both systems. However, I also find it necessary to discuss the effects of time evolution on the temperature profiles and of wind.

**Page 19**, line 296: what is the standard deviation of the temperature increase?
Line 297: 'In this case' is awkward.

Lines 298-300: The comparison with the RS is good for an average of one-hour and you well explain that such an average is necessary to mitigate the errors due to the lack of the vertical velocity measurement. I agree that there are several reasons that could explain the RASS standard deviation during one hour, but that should not affect the average (unless surface covers or advection processes would be drastically different between the RS launching site and the RASS site). So what do you mean by 'degrading the statistics'? Increasing the standard deviation?

**Pages 19 and 20 and Fig. 8**:
Are the 1-hour profiles of the RASS centered around 30 min after the RS launch (8h30 in the caption of Fig. 8)? Why did not you center them on the launch time since you are interested in the low layers?

I do not find that the 1min-RASS profiles in Fig. 8 are closer to the RS profile than the RASS 1-hour average. It is true for some points, but not for all and it is not surprising: I think it is not significant to compare the 1-min RASS profiles to the RS profile since 1-min RASS estimations include fluctuations (temperature fluctuations and fluctuations linked to the measurement process). 1-min is too short to include enough time (or space) scales of turbulence. So the 1-min profiles reveal some snapshots, which could be very different, 2 min later and a fortiori, different from a RS profile, measured 400-m apart (in addition to the fact that the radiosonde takes around 3.25 min to reach 1300 m). It could be interesting to check the time evolution of the RASS profiles during the hour. Would this evolution seem erratic like in a turbulence process? Or could we see a slow increase of the inversion layer? (I do not require that you include this in your next manuscript, but it could be interesting to discuss this point). As far as the inversion levels are concerned, I tried to join the points of the 1-min profiles in Fig. 8. The first inversion level that is detected at around 650 m with the RS is found at around 520 m and 500 m by the acoustic and parametric speakers respectively. This discrepancy could perhaps be decreased by a range correction, but there again, you could not conclude due to the lack of vertical velocity information (that can hardly be neglected when two successive 1-min profiles are compared).

Nonetheless I agree with your conclusion p. 20: 'a comparison with measurements that have both small spatial difference and high time resolution is needed to evaluate the PAA-RASS measurements.'

**Page 22**, lines 341 to 345: I do not agree with these arguments since they describe atmospheric phenomena whose time scales are larger than 1 min. I think, as said before, that the variability comes

from the local turbulence and also perhaps from the physics of measurement that is probably different between the 2 systems. Could the difference in the beam widths play a role?

The fact that the standard deviation (0.43) in the 1-min statistics of PAA vs acoustic is around the same as the standard deviation (0.4) of the 1-hour statistics of both RASS vs RS reinforces the idea that small scale processes are responsible for the variability (in addition to processes at larger scale like the increase of the inversion levels with z and time).

**Page 23**, line 364: see my questions about Fig. 6 (p. 15)
Line 368-369: I do not understand:' On the other hand, the results also suggest that the reason may include the effect of wind aloft (e.g. Fig. 5a)' since the height coverage is the same for this specific case.

Line 369: 'wind aloft' should be defined here and not p. 24, lines 364-365. Are you sure wind is measured at 1 m? (1 to 1200 m AGL). How did you compute the standard deviation of the wind?

**Page 24**, line 384: the linear regression for the acoustic speaker is not relevant. I would remove it from Fig. 10.

Lines 388-389: 'This suggests that the acoustic speaker RASS **keeps on observing** at a high altitude **even** in relatively high wind conditions.'

**Page 27**, line 423: 'may be distorted' → 'may have been distorted'.
Line 424 : 'may be needed to be steered' → might have been steered'.

**Page 29**, line 455: 'does' should be removed.

**Page 30**, line 473: I do not understand 'but frequency is likely to be less'.

Additional question: does an emission around 40 kHz require an authorization?

As a conclusion, I would be pleased to see your future works with this system, with more favorable conditions such as: the possibility to measure w, some range corrections, measurements during the whole day. I am still wondering whether the parametric system would be more efficient than the acoustic system at inversion layers (due to its narrower bandwidth).

**Additional suggestions for tables and figures:**

**Table 3**: I would use U instead of W (even if W is not w) and would add '(1 to 1200 m AGL)' after 'wind aloft'.

**Figure 5**: 'except for the first range' → '**(**except for the first range**)**'.

**Figure 6**: 'from 08:30 JST' is not convenient for the 4 profiles. See also my previous remark about Fig. 8, concerning the center of the 1-hour time average.
'The error bars represent 2 σ in the RASS **hourly** observations' ('**hourly**' added).

**Figure 7**: I would say RASS vs. radiosonde instead of radiosonde vs. RASS, but the editor will confirm or refute.

**Figure 8**:
'RASS with acoustic speakers (red) **between 8h01 and 8h02**'
'RASS with parametric speaker (blue) **between 8h02 and 8h03**'

**Figure 9**:
Same remark as in Fig. 7 for the use of vs.
'**(**except for the first gate**)**'
'a normalized frequency diagram **(color scale)**'
'the mean **hourly** data were plotted in (b)'

**Figure 10**:
'wind speed aloft **(1-1200m)**'
Please remove the linear regression for the acoustic speaker or present two legs: one horizontal from 1.5 to 5.5 m/s and another oblique one for stronger winds.

---

## Referee Comment (RC2) · Anonymous Referee #2 · 7 Jun 2019

General Comments This paper discusses operation of a Parametric Acoustic Array (PAA) speaker along with an L-band Wind profiler resulting in a Radio Acoustic Sounding System (RASS) to obtain height profiles of atmospheric virtual temperature. RASS consists of a wind profiling radar and collocated acoustic sources. Acoustic excitation is generated at a wavelength that is half of the radar transmission wavelength, so as to obtain echoes by means of Bragg scatter from the propagating acoustic wavefronts. As the sound speed generally decreases with altitude in the troposphere, the frequency of excitation required for obtaining radar backscatter at different altitudes is different in order to keep the wavelength relationship with the radar transmission. Therefore, an FM-chirped acoustic pulse is normally employed to expand the height coverage of RASS observations.

[Figure]

The acoustic sources employed for RASS work in the audible frequency range and have wide beamwidths resulting in surroundings being exposed to high acoustic levels. The main impediment to operational use of RASS has been high noise pollution in the neighbourhood of the site of its operation. Therefore, RASS with acoustic sources having low sidelobe levels is highly desirable. The application of high directivity acoustic source achieved through the non-linear interaction of ultrasonic sound waves in air resulting in audio frequencies has been exploited towards this goal.

The PAA developed by the authors generates ultrasonic frequencie at ∼37 kHz and 40 kHz. These two ultrasonic waves interact in the air and generate sum and difference frequencies. The difference frequency falls in the audible region of the acoustic spectrum. Intercomparison of measurements of virtual temperature made with PAA-RASS, conventional acoustic speaker-RASS and in situ measurements made with a GPS Radiosonde have been presented. Shortcomings of the RASS with PAA as compared to conventional acoustic source have also been outlined.

This paper presents an important step forward towards operational use of RASS.

Specific Comments

Line 54 : In order to get the true acoustic speed in a particular antenna beam direction, the radial wind speed should be subtracted from the measured acoustic speed. Therefore, "vertical wind" should be replaced with "radial wind".

Lines 138-144 : The sound pressure level (SPL) output of the PAA in the audible range is given in dB. It would be better to give the reference to weighting curve e.g. dBA or dBZ to make it more explicit. In figure 2, the SPL is measured at a distance of 25 m from the PAA. As the general practice of measuring SPL for an acoustic source is at 1 m above the source, the reason for measurement having been done at 25 m should be explained.

Why is the SPL of PAA not available in the elevation angle range of $0° - 40°$. This

measurement is of high relevance as the unique advantage of PAA is high directivity (meaning low transmission along the horizon when transmitting vertically). Effort should be made to provide these measurements.

Section 3.1 : Lines 175-177 : It is stated that "the PAA radiates bifrequency primary waves that are around 37 kHz and 40 kHz". However, Table 2 indicates Amplitude Modulation (DSB). It is not clear if these two frequencies were generated simultaneously by two halves of the PAA or the 40 kHz was modulated with the desired audio frequency. This should be clarified. Further, it is stated in line 122 that pseudorandom frequencies were chosen. What is the range of frequencies and how were they sequenced.

The ultrasonic SPL generated by the PAA is 200 dB which is extremely high. As per several studies (cf.[1] and [2]) physiological effects start manifesting in small animals at 120 dB and increase in severity with increasing SPL; exposure above 180 dB, death of a human could occur. Observations of insect, animal or bird mortality in the vicinity of the PAA should also be mentioned. Instances of hearing loss or any other discomfort faced by operators exposed to the PAA should be mentioned for the benefit of prospective users. In view of the high potential for biological hazard from this speaker, the paper should clearly mention the potential for harm from these high levels of ultrasound and give references to internationally accepted safety procedures to be adopted while using high power ultrasonic sources.

Section 4.3 : The effect of horizontal wind on the height coverage can be estimated using acoustic ray tracing. Therefore, it is recommended that the discussion about height coverage should be given with reference to the ray tracing results.

Line 408 : How was the power decreased by 15 dB – by reducing the input drive or by using smaller aperture. This clarification should be added.

Minor corrections Line 80: Replace "is expected" with "would be ideal". Line 86 : Replace "audible frequencies" with "frequencies in the audible range". Line 88 : Replace "Hence after" with "Thereafter". Line 107 : Add "and" between Oceanic and

Atmospheric Line 124 : Replace "comprised" with "consists of". Lne 136 : Replace "broaden" with "broadened". Line 199 : Replace "reached" with "were obtained from altitudes" Line 200 : Replace "also reached" with "were obtained from" Lines 396- 399 : The sentence need to be rewritten. I suggest as follows, "Since the four acoustic speakers were not adjusted in phase, this robustness could be explained by the higher aggregate sound power than that shown in Fig. 2 and possible location of sound waves above the antenna in spite of relatively high winds." Line 429 : Replace "availability" with "applicability".

References [1]. Environmental Health Criteria 22 : Ultrasound, World Health Organisation Geneva 1982. [2]. Guidelines for the Safe Use of Ultrasound: Part II _ Industrial and Commercial Applications Safety Code 24, Health Canada, 1991.

Please also note the supplement to this comment:
https://www.atmos-meas-tech-discuss.net/amt-2019-92/amt-2019-92-RC2-supplement.pdf

---

## Author Comment (AC1) · 1 Aug 2019

**Response to Reviewer 1**

We are grateful to the reviewer for the very careful and thorough examination of our manuscript. We think the revised manuscript is substantially improved as a result.

Specific Comments:

1-1. Page 2, line 24: I do not understand the use of 'if' in this sentence.

1-2. Line 29: I would write 'versus' instead of '/' in 'RASS/radiosonde'.

**Response:**

1-1. We have revised the manuscript and eliminated the text beginning with "if" (line 24).

1-2. Done (line 28).

2-1. Page 3, line 47: 'measures' => I suggest 'provides'

2-2. Line 48: I would add 'from the measurement of the Doppler shift between the emitted and backscattered radar wave', after 'propagates vertically'

2-3. Line 49: 'among' => 'between'

**Response:**

2-1. Done (line 46).

2-2. We have added 'from the measurement of the Doppler spectrum" in lines 47–48 to make it consistent with the Doppler spectrum in Fig. 4.

2-3. Done (line 49).

3-1. Page 4, lines (54-55): This issue about the vertical velocity correction is very important and not well presented. It is obvious that w has to be included. The fact that you could not use w is an issue, and you show in the following how you cope with it (1-hour averages). The way the issue is introduced, at the beginning of the paper, is not satisfying especially the sentence : 'the vertical wind speed is considered in Eq (1) because the neglect of the wind velocity ....'. I explain why.

At the beginning (p 4), it is obvious that the correction has to be done. You mention p 7 (lines 112- 115), that w could not be used due to the automobiles echo. So I deduced you had used the oblique beams instead of the vertical one. I understood in the following that you did not. Please, could you clarify this issue in your presentation, perhaps by saying at the beginning that you had to neglect w and that this issue would have to be discussed?

3-2. If you cite May's and Angevine's papers it could be interesting to add (further on in the manuscript) some details of their discussion, including the discussion on the downward bias in the profiler velocity measurements mentioned in Angevine et al., 1998: the 1-hour average is a way to mitigate this effect.

**Response:**

3-1. We have added an explanation for this comment (lines 56–59 and 136–138): "However, we do not consider the radial wind speed in our experiments, because strong clutter sometimes contaminates the Doppler spectrum and masks the atmospheric echo in the vertical beam observation. This issue is addressed in later sections.", and "Indeed, the vertical velocity correction can decrease the accuracy of RASS in situations with calm wind and a lower reliability of vertical wind measurements (Görsdorf and Lehmann 2000)."

3-2. We have modified the manuscript to respond to this comment (lines 209–211): "Using the hourly mean may also reduce the daytime downward bias (e.g., Görsdorf and Lehmann 2000; Adachi et al., 2005), which could be attributable to insects or hydrometeors that are undetectable (Angevine, 1997)."

4-1. Line 63: I would add 'accurate' before 'corrections'

4-2. Line 64: if relevant, I would add 'thermodynamic' before 'constants'

**Response:**

4-1. Done (Line 82).

4-2. Done (Line 83).

5. Minor comment: in lines 57, 59 and 61 respectively, you use brackets for two different purposes: with 'bias (systematic error)', 'standard deviation (precision)', the brackets provide the meanings whereas in 'bias (standard deviation)', you present an alternative. The latter use of the brackets is continued through the manuscript. The first use could be presented in a different way.

**Response:** We have revised out text to respond to this comment (lines 60–62): "The systematic error or bias of the virtual temperature measurements", and "while the standard deviation or precision has also been reported around 1°C.".

6-1. Lines 61 to 64: I was not convinced by the results you cite (Görsdorf's and Lehman's work) as long as I had not read the paper. My reservation was about the one year time average that reduced the bias as expected. When reading this paper, I was convinced and I found additional interesting information that would deserve to be discussed in your own paper (but probably not in the introduction):

- the reason for the range correction would be helpful to comment (see my point, Page 16, line 245). Görsdorf and Lehman are not the only ones who mention this range correction: Angevine et al., 1998 also do, but in the references you cite, Görsdorf and Lehman discuss the benefit of the correction at the inversion levels, which is a major advance and could feed your discussion.

- the distinction between climatological investigation and individual profile comparison is also an important point to comment: you can hardly compare errors amounts when they are computed with

varying time-scales (unless you discuss the conditions).

To conclude, I would say that the use of this reference comes too early and that you could take more advantage in discussing it further on.

6.2. In general, I suggest that you give more details about the results and the uncertainties that are reported in the literature, concerning the RASS assessment. For instance, Angevine et al., 1998 made a comparison of Tv during 1 month, at one level (396m agl) between a RASS system (UHF) and tower instruments. The uncertainties that they reported are expected to be different from those obtained for instance by Moran and Strauch (1994) who used a RASS with a VHF profiler, during 5 weeks (400 profiles, compared to RS). Moreover, in each case, some information should be given about the corrections (vertical velocity correction, range correction ...).

**Response:**

6-1. We appreciate the reviewer's suggestion. However, we did not find any discussion on the benefit of the range correction at the inversion levels in Görsdorf and Lehmann (2000). Instead, we have added an explanation to respond to the comment (lines 387–388): "the discrepancies in and below the inversion could be mitigated by considering the effect of the vertical airflow and/or **applying a range correction** "

6-2. We have revised the manuscript to refer more details reported in the literature and respond this comment (line 62-83): "May et al. (1989) compared..."

7. Page 5, line 66: you could also add 'the distortion of the acoustic wave due to turbulence and vertical temperature gradients (Lataitis, 1992)'

  **Response:** We have revised the text to respond to the comment (lines 86–87): "…in addition to the effects of **turbulence and vertical temperature gradients** (Lataitis, 1992)."

8. Line 68 : 'A wind profiler/RASS has been used'=>Joined measurements of wind profiler and RASS have been frequently used'.

  **Response:** We have revised the text to respond to this comment (line 88): "A wind profiler with RASS has been frequently used"

9. Line 71: 'Important limitations of this method include the emission' => 'Among the limitations of this method, an important one is the emission'

  **Response:** Done (lines 91–92).

10. Line 80 : 'is expected' => 'would be promising'

  **Response:** We have replaced "is expected" with "would be ideal" in line 101.

11-1. Page 7, line 100: please provide the latitude of the site

11-2. Line 106: 'The MRI wind profiler' => 'The MRI 1357-MHz wind profiler'

11-3. Line 113: see my previous remarks about w (p. 4)

**Response:**

11-1. We have added the latitude of the site in Table 1.

11-2. We added the frequency in the manuscript (lines 128–129): "The profiler used in this study operated at 1357.5 MHz with 100 m pulse lengths".

11-3. Please see our response #3-1.

12-1. Page 8, line 119: 'In the experiments' => 'For the experiment purpose'

12-2. Line 122-123: about pseudorandom frequencies. This is either too much or not enough explained for a non-expert reader. Is it necessary to mention? I suggest to move this specificity to Table 1.

**Response:**

12-1. Done (line 142).

12-2. We have copied this term to Table 1 but also added an explanation for this term to respond to the comment (lines 150–152): "The frequency sweeps were randomly shuffled within each frequency range to make acoustic spectrum almost uniform (Angevine et al., 1994)".

13-1. Page 9, line 134-137: am I right to say that the beamwidth of the speaker should be larger than the MRI beamwidth? Does that mean that you start with degraded conditions with the default sound beam? Is it the point you raised when you spoke of the parametric speaker directivity (line 22, page 2)? I raise this point since I wondered at the beginning whether the directivity would be an advantage or a drawback.

13-2. Line 137: turbulence will broaden the sound beam width, especially inside the boundary layer, but it will also reduce the measurement range, as you said before.

13-3. Line 137: 'aloft' is not accurate enough and I wonder whether it is appropriate here.

**Response:**

13-1. We believe that the beam width of the speaker should match that of profiler, because if the former is much narrower than the latter, no RASS echo is observed. On the other hand, if the former is larger than the latter, the height coverage is reduced because of decrease of the peak power as shown in Fig 2. We have added an explanation to respond to this comment (lines 162-168): "Prior to the designing of the MRI PAA, we made…".

13-2. The reviewer is right. However, we expect that the size of the sound spot matches the width of the radio wave width by the beam broadening due to turbulence. We have added an explanation to respond to the comment (line 172): "and match the radio beam width".

13-3. We have removed "aloft" (line 171).

14-1. Line 138: 'measured in the field' I would add: 'with a sound level meter'.

14-2. Could you describe the measurement protocol?

**Response:**

14-1. Done (line 175).

14-2. We have an explanation of the protocol to respond to this comment (lines 173-180): "In order to measure the audible sound pressure level…" .

15. Line 143: 'and is **therefore** significantly more annoying' (therefore can be added if you find it relevant).

**Response:** We have added the term in line 186.

16. Page 11, line 161: at 10-m height?

  **Response:** This was typo as the reviewer pointed out, and we have revised the manuscript to respond to this comment (lines 203–204): "at 20 m AGL".

17-1. Lines 163-167 : my first reaction is that the launching occurred in a period of the day when the time temperature gradient is important and the boundary layer top level is rapidly increasing. You discuss this point later on, but you could already add a few comments.

17-2. I also suggest to add a column in Table 3, with the time difference between the launching and the sunrise time. I calculated 2h30 to 4h. Am I right?

**Response:**

17-1. We have revised the manuscript to respond to this comment (line 205): "which may be preferable for the formation of inversion layer".

17-2. Since the analysis of the inversion layer evolution is beyond the scope of the present study, we did not provide the sunrise time for all the experiments in Table 3 but added it just for a sample profile for Fig. 8 (line 372): "This observation was made more than 3 h after sunrise (05:15 JST) on that day." Yes, the reviewer's estimation time was right.

18. Line 170: I do not understand the meaning of 'availability'. Do you mean 'practical relevance'?

**Response:** We have replaced "availability" with "applicability" (line 215).

19. Page 12, line 182: tell me if I am wrong, but it seems to me that the reference (ISO 1993) is not easily available. Could you provide (in an appendix) the equations to calculate the sound attenuation coefficient and the attenuation, derived from the RS measurements?

  **Response:** The reference is available online at: https://www.iso.org/standard/17426.html. However,

we have added a derivation of equations and some results in an appendix to respond to this comment.

20-1. Page 13, line 200: 'the PAA also reached a minimum altitude of 1.1 km AGL' => 'the PAA reached an altitude of 1.1 km AGL'.

20-2. Lines 203-206: this unique example is not enough to draw a conclusion. So, I would use 'may reach' instead of 'can reach' and indicate that the propagation level will be systematically compared between the 2 systems. or, you can simply say that the example in Fig. 4 is promising....

**Response:**

20-1: Done (line 246).

20-2: We have replaced "can reach" with "may reach" in line 250, and added "The height coverage of the two speaker systems are discussed later." in lines 252–253 to respond to this comment.

21. Fig. 5: usually, the backscattered signal of wind profiler radars is artificially corrected at the first gate to avoid receiver saturation. Are you sure it is not the case for RASS systems? Could this be a reason for the increase of the echo power between the first and the second gate?

**Response:** We think that the reviewer is talking about the STC. Our wind profiler does not implement the STC since it is equipped with an A/D converter with wide (16-bit: 96dB) dynamic range. We believe the increase between the first and the second gate were caused by other reasons, as Lataitis (1992) pointed out. We have added an explanation to respond to this comment (lines 311–314): "factors including the recovery of the receiver and incomplete overlapping of the electromagnetic and acoustic beams due to the special separation between the antenna and speaker systems could lead to a significant gradient in the receiving power at this gate (Lataitis, 1992)."

22-1. Page 15, line 228: 'available' => 'convenient'

22-2. Line 231: 'immediately'=> 'sharply'

**Response:**

22-1. We have replaced "available" with "applicable" in line 287.

22-2. Done (line 290).

23-1. Fig. 6: I suppose your sample of profiles is the best you could provide. Would it be possible to have a look on the corresponding Received Power profiles?

23-2. Why is there such a large difference in the number of data (320 and 237 in Fig. 6a) while the measurement range is the same for both systems? On the opposite, in Fig. 6d, 96-90=6 is small while the PAA vertical range is 200 m lower than the acoustic speaker. I probably miss something.

23-3. I suppose you tried comparisons with RS data averaged during 15 min or 30 min. Were the results so bad?

**Response:**

23-1. We have added profiles of $P_r$ in Fig. 6, and an explanation to respond to this comment (lines 317–321): "It is noteworthy …a received RASS echo power…".

23-2. This happens because the height coverage changes with time especially for the PAA as shown error bars in original Fig. 10. In that figure, the second left (2.1m/s) is the data for Fig. 6a. (see Table 3). The PAA has longer error bar than the ACS, showing wider variation of the height coverage. In contrast, for Fig. 6d (4[th] left, 3.1m/s in original Fig. 10), the error bars are relatively short for each system indicating small variation of the coverage. The height coverage in Fig. 6 reflects the maximum height coverage in the RASS duration. Therefore, the number of the data has weak relation with the height coverage; rather, it is related to mean height coverage as written in lines 348–349. To avoid confusion, we have revised the manuscript (lines 438–439 and 461–464): "which (is =>) may also be reflected by the fewer number of data in the statistics", and "Note that the mean RASS height coverage shown in the figure is different from the height coverage of the mean virtual temperature profile in Fig. 6, because the latter reflects the maximum height coverage within the observed profiles after quality control in the duration of the RASS measurement."

23-3. We have tried to average for 30 min from 0815 to 0845. For instance, on 15 October, when there was no inversion layer aloft, the statistical results for 30 min mean, (bias, STD, N) = ACS(-0.19, 0.13, 165), and PAA(-0.18, 014, 124) were not very different from those for 60 min in bias (Fig. 6a). However, in the case of 8 November, when inversions were evident, the statistics, ACS(-0.13, 0.26, 152) and PAA(-0.81, 0.32, 106), became worth in bias than that for 60 min (Fig. 6c). Standard deviations were slightly improved by averaging over 30 min for both cases because it may reflect the time evolution of the temperature profile. However, the maximum height coverage for PAA reduced 100 m (one gate) by decreasing mean time for both cases although those for the ACS did not change.

24-1. Page 16, line 245: '(e.g. Figs 6n and 6c)' => 6a also.

24-2. Line 249 : I suggest that you give here the reason for the range corrections presented by Görsdorf and Lehmann and that you briefly explain the processing.

24-3. The range corrections should be applied to the whole profile (and not only at the first gate), to mitigate the discrepancies at the inversion levels.

**Response:**

24-1. Done (line 309).

24-2. We have added an explanation to respond to the comment (lines 314–317): "In addition, a range error (e.g., Angevine and Ecklund, 1994; Görsdorf and Lehmann, 2000; Johnston et al., 2002) caused by the height variation of the backscatter intensity may also contribute to the smaller received power."

24-3. We have revised the text as mentioned above (#24-2) and in lines 387–388: "… the discrepancies in and below the inversion could be mitigated by … **applying a range correction**.

25. Lines 251 to 253 (page 16, 'The RASS data were averaged for about an hour. The RS data were smoothed by 100 m running means to match the RASS observations)' should be moved p15, just before the comments on Fig. 6.

**Response:** Done (line 301).

26-1. Page 18, line 278: as said before for Fig. 6, I'm not convinced by the argument of the link between the smaller data number for the PAA and the height coverage. In addition, the wind is relatively stronger at 1300m in Fig. 5a, while the height coverage is similar for both systems.

26-2. However, I also find it necessary to discuss the effects of time evolution on the temperature profiles and of wind.

**Response:**

26-1. Please see our response #23-2.

26-2. We have added an explanation to respond to this comment (lines 464–466): "The long error bars may reflect the large time evolution of the RASS height coverage, which may also be related to the evolution of the wind in the duration."

27-1. Page 19, line 296: what is the standard deviation of the temperature increase?

27-2. Line 297: 'In this case' is awkward.

**Response:**

27-1. We have added "with a standard deviation of 1.0°C" in lines 366–367.

27-2. We have replaced " In this case" with "Thus" in line 367.

28. Lines 298-300: The comparison with the RS is good for an average of one-hour and you well explain that such an average is necessary to mitigate the errors due to the lack of the vertical velocity measurement. I agree that there are several reasons that could explain the RASS standard deviation during one hour, but that should not affect the average (unless surface covers or advection processes would be drastically different between the RS launching site and the RASS site). So what do you mean by 'degrading the statistics'? Increasing the standard deviation?

**Response:** We mean that an average period may affect not only the standard deviation but also the bias by "degrading statistics". For instance, 10 min mean Tv [°C] observed at 1.2 m AGL from 08:00 JST for 1 hour on 15 October 2016 were as follows, (08:00, 14.175), (08:10, 14.900), (8:20, 15.840), (8:30, 17.185), (8:40, 17.158), (8:50, 17.572), and (9:00, 18.142). The mean for this hour is 16.42°C at 08:30 but this value may change with averaging period (Note 17.19°C was observed at 08:30). Although they are values observed near the ground, we believe that an average period may affect the bias and would like to retain "degrading the statistics" in the text (line 370).

29. Pages 19 and 20 and Fig. 8:

Are the 1-hour profiles of the RASS centered around 30 min after the RS launch (8h30 in the caption of

Fig. 8)? Why did not you center them on the launch time since you are interested in the low layers?

**Response:** We centered the RASS duration on the launch time; The RSs were launched not at 08:00 JST

but at 08:30 JST as written in line 201, and the most of RASS durations were from 08:01 to 09:00 to

center the RS launch as shown in Table 3.

30-1. I do not find that the 1min-RASS profiles in Fig. 8 are closer to the RS profile than the RASS

1-hour average. It is true for some points, but not for all and it is not surprising:

30-2.  I think it is not significant to compare the 1-min RASS profiles to the RS profile since 1-min

RASS estimations include fluctuations (temperature fluctuations and fluctuations linked to the

measurement process). 1-min is too short to include enough time (or space) scales of turbulence. So

the 1-min profiles reveal some snapshots, which could be very different, 2 min later and a fortiori,

different from a RS profile, measured 400-m apart (in addition to the fact that the radiosonde takes

around 3.25 min to reach 1300 m).

30-3. It could be interesting to check the time evolution of the RASS profiles during the hour. Would this

evolution seem erratic like in a turbulence process? Or could we see a slow increase of the inversion

layer? (I do not require that you include this in your next manuscript, but it could be interesting to

discuss this point).

30-4. As far as the inversion levels are concerned, I tried to join the points of the 1-min profiles in Fig. 8.

The first inversion level that is detected at around 650 m with the RS is found at around 520 m and

500 m by the acoustic and parametric speakers respectively. This discrepancy could perhaps be

decreased by a range correction, but there again, you could not conclude due to the lack of vertical

velocity information (that can hardly be neglected when two successive 1-min profiles are compared).

**Response:**

30-1. We have revised the text to respond to the comment (lines 378–380): "By contrast, the 1 min raw

RASS data recorded around the radiosonde launch time represented the inversion layer better than the

mean RASS measurements **at some points**".

30-2. As reviewer pointed out, 1-min is too short to include the time and space scales of turbulence and

causes large error, if the temperature fluctuation due to turbulence is very large. However, in that case,

the radiosonde data sampled at 1 s cannot be used as a standard reference temperature measurement

because the mean time for radiosonde observation at each height is less than 17 s even after applying

100 m running means. Since there are some conformities between the radiosonde and RASS in

measuring Tv profiles, we believe (and assume) the temperature fluctuation due to fluctuation was

small and averaged out to some extent. We have revised the text to take the effect of turbulence into

account and to respond to this comment (lines 303–304 and 380–382): "The running mean may also play a role to mitigate the effect of the temperature fluctuation due to turbulence on the radiosonde measurements.", and "which may have been due to the locality of the inversion layer, the effects of vertical air motion or **turbulence**, ...".

30-3. We have tried to make a time-height cross section of $Tv$ (Fig. 1R) just for the reviewer. Although a cold layer, which was a component of the inversion layer, was analyzed at about 600 m AGL from 08:05 JST (or before), the layer is evanescent in the figure. On the other hand, since the temperature observed at surface did not fluctuate as is seen at low altitudes (200 – 400 m AGL) in this figure, this fluctuation was not true but was likely biased by the vertical air flow. Thus, we expect that a long-lived cold layer would be analyzed longer in time if the correction for the vertical component of the wind can be applied. However, since it is beyond the scope of this study, we do not think it is interesting to discuss the evolution of the inversion layer with this figure.

30-4. We have added an explanation to respond to the comment (lines 387–388): "the discrepancies in and below the inversion could be mitigated by considering the effect of the vertical airflow and/or **applying a range correction**."

31. Nonetheless I agree with your conclusion p. 20: 'a comparison with measurements that have both small spatial difference and high time resolution is needed to evaluate the PAA-RASS measurements.'
**Response:** We appreciate the reviewer's comment on this.

32. Page 22, lines 341 to 345: I do not agree with these arguments since they describe atmospheric phenomena whose time scales are larger than 1 min. I think, as said before, that the variability comes from the local turbulence and also perhaps from the physics of measurement that is probably different between the 2 systems. Could the difference in the beam widths play a role? The fact that the standard deviation (0.43) in the 1-min statistics of PAA vs acoustic is around the same as the standard deviation (0.4) of the 1-hour statistics of both RASS vs RS reinforces the idea that small scale processes are responsible for the variability (in addition to processes at larger scale like the increase of the inversion levels with z and time).
**Response:** We agree that the standard deviation (0.43) in the 1-min statistics of PAA vs. acoustic could be mostly due to local turbulence since the spatial difference was negligible and the time difference was quite small as the reviewer pointed out. We have revised the text to respond to this comment (lines 415–417 and 564–565): "Since the spatial difference was negligible and the time difference was quite small, the reason for this discrepancy could include **temperature fluctuation due to turbulence**.", and "temperature fluctuation due to turbulence could contribute to deteriorate the statistics." We do not think that the difference in the beam width play the role (as long as temperature is horizontally homogeneous, which we assumed in comparisons with radiosondes).

33-1. Page 23, line 364: see my questions about Fig. 6 (p. 15)

33-2. Line 368-369: I do not understand:' On the other hand, the results also suggest that the reason may include the effect of wind aloft (e.g. Fig. 5a)' since the height coverage is the same for this specific case.

33-3. Line 369: 'wind aloft' should be defined here and not p. 24, lines 364-365.

33-4. Are you sure wind is measured at 1 m? (1 to 1200 m AGL). How did you compute the standard deviation of the wind?

**Response:**

33-1. Please see our response # 23-2.

33-2. See our response # 23-2. We have also revised the manuscript to avoid confusion (lines 293-295): "This suggests that the PAA **has enough peak power to reach the highest range gate but** is more susceptible to high winds than the acoustic speakers".

33-3. We would retain the location of the definition of "wind aloft" in line 456, because this range is defined just for Fig. 10, which is introduced first in line 452.

33-4. This was a typo, and we have corrected (lines 456–457): "the mean wind from **20** to 1200 m AGL (Table 3)". The standard deviation of the wind speed in Table 3 was derived from the radiosonde data by

$$\sigma = \sqrt{\frac{1}{N}\sum_{i=1}^{N}(x_i - \bar{x})^2}$$

, where $x_i$ is the observed $i^{th}$ wind speed data from the beginning of the observation, and the altitude of the $N^{th}$ data is the closest to but less than or equal to 1200 m AGL.

34. Page 24, line 384: the linear regression for the acoustic speaker is not relevant. I would remove it from Fig. 10.

**Response:** We have revised Fig. 10 based on the reviewer's comment # 47-2.

35. Lines 388-389: 'This suggests that the acoustic speaker RASS **keeps on observing** at a high altitude **even** in relatively high wind conditions.'

**Response:** Done (line 477).

36-1. Page 27, line 423: 'may be distorted' => 'may have been distorted'.

36-2. Line 424 : 'may be needed to be steered' => might have been steered'.

**Response:**

36-1. Done (lines 515–516).

36-2. Done (line 516).

37. Page 29, line 455: 'does' should be removed.

**Response:** Done (line 574).

38. Page 30, line 473: I do not understand 'but frequency is likely to be less'.

**Response:** This term (line 592) means the frequency of the occurrence in other seasons, not the frequency of the acoustic and/or radio wave.

39. Additional question: does an emission around 40 kHz require an authorization?

**Response:** There is no requirement for authorization at least in Japan. However, there are some recommendations from organizations including WHO. We have added Section 4.4 to respond to this comment.

40. As a conclusion, I would be pleased to see your future works with this system, with more favorable conditions such as: the possibility to measure w, some range corrections, measurements during the whole day. I am still wondering whether the parametric system would be more efficient than the acoustic system at inversion layers (due to its narrower bandwidth).

**Response:** We thank the reviewer for his/her positive conclusion.

41. Table 3: I would use U instead of W (even if W is not w) and would add '(1 to 1200 m AGL)' after 'wind aloft'.

**Response:** Done (Table 3).

42. Figure 5: 'except for the first range' => '(except for the first range)'.

**Response:** Done (Fig. 5).

43-1. Figure 6: 'from 08:30 JST' is not convenient for the 4 profiles. See also my previous remark about Fig. 8, concerning the center of the 1-hour time average.

43-2. 'The error bars represent 2 σ in the RASS hourly observations' ('hourly' added).

**Response:**

43-1. Please see our comment #29; The RSs were launched not at 08:00 JST but at 08:30 JST in Japan.

43-2. Done (Fig. 6).

44. Figure 7: I would say RASS vs. radiosonde instead of radiosonde vs. RASS, but the editor will confirm or refute.

**Response:** Done (Fig.7).

45. Figure 8:'RASS with acoustic speakers (red) between 8h01 and 8h02' 'RASS with parametric speaker (blue) between 8h02 and 8h03'

**Response:** Since radiosonde was launched at 08:30, we did not change the time for the RASS measurement. Instead, we have revised the text to make it clear (Fig. 8): "closed circles represent 1 min raw data **from the time indicated**."

46-1. Figure 9: Same remark as in Fig. 7 for the use of vs.

46-2. '(except for the first gate)'

46-3. 'a normalized frequency diagram (**color scale**)'

46-4. 'the mean **hourly** data were plotted in (b)'

**Response:**

46-1– 46-3. Done (Fig. 9).

46-4. We have replaced "mean data" with "hourly-mean data (Fig. 9).

47-1. Figure 10: 'wind speed aloft **(1-1200m)**'

47-2. Please remove the linear regression for the acoustic speaker or present two legs: one horizontal from 1.5 to 5.5 m/s and another oblique one for stronger winds.

**Response:**

47-1. We have replaced "aloft" with "aloft (20–1200 m AGL)" in Fig. 10.

47-2. Done in Fig. 10. However, we have replaced the mean wind speed aloft with the horizontal displacement for the horizontal axis.

[Figure]

Fig. 1R. Time height cross section of virtual temperature obseved with RASS averaged over 2 min.

---

## Author Comment (AC2) · 1 Aug 2019

**Response to Reviewer 2**

We are grateful to the reviewer for the very careful and thorough examination of our manuscript. We think the revised manuscript is substantially improved as a result.

Specific Comments:

1.  Line 54: In order to get the true acoustic speed in a particular antenna beam direction, the radial wind speed should be subtracted from the measured acoustic speed. Therefore, "vertical wind" should be replaced with "radial wind".

**Response:** We have replaced "vertical wind" with "radial wind" in lines 50, 54 and 56.

2. Lines 138-144:

2-1. The sound pressure level (SPL) output of the PAA in the audible range is given in dB. It would be better to give the reference to weighting curve e.g. dBA or dBZ to make it more explicit.

2-2. In figure 2, the SPL is measured at a distance of 25 m from the PAA. As the general practice of measuring SPL for an acoustic source is at 1 m above the source, the reason for measurement having been done at 25 m should be explained.

2-3. Why is the SPL of PAA not available in the elevation angle range of 0° − 40°. This measurement is of high relevance as the unique advantage of PAA is high directivity (meaning low transmission along the horizon when transmitting vertically). Effort should be made to provide these measurements.

**Response:**

2-1. We have added "dBA" (line182): "the SPL was less than 55 dB (**dBA**) at a zenith angle of 40°"

2-2. We have added an explanation to respond to this comment (line 174): "The measurements were made … at a distance of 25 m, because a range of 10 m would be necessary to complete producing audible sound from ultrasound with a PAA of this size (Prof. Kamakura, 2018, personal communication). Safety was also considered for the level-meter operators in determining the distance, as is discussed later."

2-3. We have revised Fig. 11 to show the SPL of the PAA.

Section 3.1 :

3. Lines 175-177: It is stated that "the PAA radiates bifrequency primary waves that are around 37 kHz and 40 kHz". However, Table 2 indicates Amplitude Modulation (DSB). It is not clear if these two frequencies were generated simultaneously by two halves of the PAA or the 40 kHz was modulated with the desired audio frequency. This should be clarified.

**Response:** The reviewer is right. The term DSB in the original Table 2 was wrong and removed. (We

don't think that the AM works well for RASS because ultrasound, the carrier, cannot propagate long distances.)

The two frequencies were generated simultaneously by all transducers of the PAA. To avoid confusion, we have revised our manuscript (lines 219–222): "The MRI PAA radiates bifrequency primary waves that are around 37 kHz and 40 kHz from all the transducers **simultaneously** to generate the parametric sound".

4. Further, it is stated in line 122 that pseudorandom frequencies were chosen. What is the range of frequencies and how were they sequenced.

**Response:** We have added an explanation for this comment (lines 145–152): "Prior to every experiment, an acoustic wave with a wide frequency range (2715 to 3265 Hz corresponding to about ±50°C) was generated to detect center Doppler frequency of the RASS echo. Then, during each experiment, the emitted acoustic frequency range was automatically narrowed down to a shorter frequency span (130 Hz, corresponding to about ±12°C) around the detected center frequency to increase SNR and height coverage. The frequency sweeps were randomly shuffled within each frequency range to make acoustic spectrum almost uniform (Angevine et al., 1994)."

5. The ultrasonic SPL generated by the PAA is 200 dB which is extremely high. As per several studies (cf.[1] and [2]) physiological effects start manifesting in small animals at 120 dB and increase in severity with increasing SPL; exposure above 180 dB, death of a human could occur. Observations of insect, animal or bird mortality in the vicinity of the PAA should also be mentioned. Instances of hearing loss or any other discomfort faced by operators exposed to the PAA should be mentioned for the benefit of prospective users. In view of the high potential for biological hazard from this speaker, the paper should clearly mention the potential for harm from these high levels of ultrasound and give references to internationally accepted safety procedures to be adopted while using high power ultrasonic sources.

**Response:** We have added Section 4.4 for this comment. We have also added some considerations for safety in the revised manuscript (e.g., line 178 and Fig. 1d). We thank the reviewer for this comment and providing the references.

6. Section 4.3: The effect of horizontal wind on the height coverage can be estimated using acoustic ray tracing. Therefore, it is recommended that the discussion about height coverage should be given with reference to the ray tracing results.

**Response:** We have estimated the horizontal displacement of the sound from the radio wave by using acoustic ray tracing and revised Figs. 5 and 10 to discuss the effect of horizontal wind on height coverage in Section 4.3.

7. Line 408: How was the power decreased by 15 dB – by reducing the input drive or by using smaller aperture. This clarification should be added.

**Response:** The decrease in line 497 was not made manually but associated with the beam broadening as written in the same line. On the other hand, the peak power was reduced manually in measuring the SPL at multiple zenith angle (Fig. 11). We have added an explanation for this comment (line 503): "The peak power was decreased by about 7.5 dB by reducing the power supply to the PAA amplifier, which decreased not only the audible sound but also the ultrasound levels for practical reasons (noisy) and measurement safety."

Minor corrections

8. Line 80: Replace "is expected" with "would be ideal".

**Response:** Done (line 101).

9-1. Line 86: Replace "audible frequencies" with "frequencies in the audible range".

9-2. Line 88 : Replace "Hence after" with "Thereafter".

**Response:**

9-1. Done (line 106).

9-2. Done (line 109)

10. Line 107: Add "and" between Oceanic and Atmospheric

**Response:** Done (line 127).

11. Line 124: Replace "comprised" with "consists of".

**Response:** Done (line 153).

12. Line 136: Replace "broaden" with "broadened".

**Response:** Done (171).

13. Line 199: Replace "reached" with "were obtained from altitudes"

**Response:** Done (line 245).

14. Line 200: Replace "also reached" with "were obtained from"

**Response:** We have modified the text to response to a comment from another reviewer (line 246): "the PAA reached an altitude of 1.1 km AGL".

15. Lines 396- 399: The sentence need to be rewritten. I suggest as follows,
"Since the four acoustic speakers were not adjusted in phase, this robustness could be explained by the higher aggregate sound power than that shown in Fig. 2 and possible location of sound waves above the antenna in spite of relatively high winds."

**Response:** Done (lines 485–488). Thank you.

16. Line 429: Replace "availability" with "applicability".

**Response:** Done (line 546).

---

## Referee Report (RR1)

**Response to author**

**Reviewer 1 (2nd review)**

Thanks for the new manuscript and for the efforts you made to answer to both reviewers. I particularly appreciated the detailed description of the results in the literature, the addition of the received power in Fig. 6 and 8, the appendix addition, the paragraph on security issues, the use of the horizontal displacement and the answers to my own remarks, including Fig.1R.

However I noted some points that still require improvements. **Your** text is written **in bold characters**, my proposal is in normal characters.

Lines 45-48
**When using RASS techniques, one or more acoustic sources are co-located with an antenna, and the profiler provides the vertical profile of the speed at which the acoustic disturbance propagates vertically (Angevine et al., 1994) from the measurement of Doppler spectrum**
➔ When using RASS techniques, one or more acoustic sources are co-located with an antenna, and the profiler (from the measurement of Doppler spectrum) provides the vertical profile of the speed at which the acoustic disturbance propagates vertically (Angevine et al., 1994).

I am still not satisfied with this: you can remove the words in brackets if you prefer.

Lines 53-59
**Thus, a vertical profile of the speed of sound can be converted to a profile of virtual temperature. The radial wind speed is considered in Eq. (1) because the neglect of the wind velocity along the beam is the largest source of error in RASS measurements (e.g., May et al., 1989; Angevine et al., 1994). However, we do not consider the radial wind speed in our experiments, because strong clutter sometimes contaminates the Doppler spectrum and masks the atmospheric echo in the vertical beam observation. This issue is addressed in later sections.**
➔ Usually experimenters use the vertical beam to minimize the wind velocity correction. The neglect of w in Eq. (1) may be a large source of error in RASS measurements (e.g. May et al., 1989; Angevine et al., 1994). In spite of this, we were not able to consider w in this work because strong clutter sometimes contaminated the Doppler spectrum and masked the atmospheric echo in the vertical beam observation. This issue is addressed in later sections.

6-1. **We appreciate the reviewer's suggestion. However, we did not find any discussion on the benefit of the range correction at the inversion levels in Görsdorf and Lehmann (2000)** : you are right. I extrapolated their results and it is a mistake.

Line 83 : **are applied** ➔ were applied

Lines 136-138 :

**Indeed, the vertical velocity correction can decrease the accuracy of RASS in situations with calm wind and a lower reliability of vertical wind measurements (Görsdorf and Lehmann 2000).**

I do not approve this sentence, that cannot be used out of context to justify that you neglected w: there is not enough indication of the wind vertical velocities under your experimental conditions, even if it is early in the morning, under anticyclonic conditions and weak horizontal wind. No geographic indication is given to discard the possibility of sea breeze or mountain breeze for instance. I would then remove the sentence. The reader will easily understand that it is better not to make the velocity correction, when the error is likely to be larger than the correction.

Line 174 : 'flame' ????

Do you mean 'a vertical frame'? This could be a reason why you cannot provide measurements at low elevation (high elevation in this position) as required by Reviewer 2 (see her/his remark 2.3). As far as this remark is concerned, I cannot see how Fig. 2 has been revised. I'm wondering whether there would not be a confusion between the 'zenith angles' you indicate (0 to 40 deg) and the 'elevation angles' that Reviewer 2 mentions (0 to 40 deg ie 50 to 90 deg zenith angles). Anyhow I understood that your system is designed to work at high elevation ...

Line 176 : **because a range of 10 m would be necessary to complete producing audible sound**
> ➔ because a range of 10 m at least is necessary to produce audible sound

Line 178 : instead of the **personal communication,** could not this be simply explained by considerations of geometry?

Line 205: **which may be preferable for the formation of the inversion layer** : I would not use 'preferable' since the quick variation of the inversion layer is more a drawback than a positive point. OK not to mention it now (and remove the highlighted text above). It is enough to mention this point and the resultant variability in the 1-hour results you show in the following.

Lines 209-211:

**Using the hourly mean may also reduce the daytime downward bias (e.g., Görsdorf and Lehmann 2000; Adachi et al., 2005)**
It would not reduce the downward bias of the vertical velocity but the effect of this bias on the data dispersion. Neither Görsdorf and Lehmann 2000 nor Adachi et al., 2005 said that 1 hour averaging would reduce the bias, although they mentioned the downward bias of w and its effect on their results (10 min- averages and 30 min, respectively). Görsdorf and Lehmann 2000 suggested not to use the vertical velocity for long-term measurements and climatological investigations, to avoid the downward bias, but said that w should be taken into account for comparisons of individual profiles. However, shorter integration times (shorter than 10 minutes) would increase the systematic error in the w estimation. You perfectly know what Adachi and al. 2005 said…

**which could be attributable to insects or hydrometeors that are undetectable (Angevine, 1997)**
➔ other factors could contribute : see for instance

Muschinski and Sullivan, 2013. Using large-eddy simulation to investigate intermittency fluxes of clear-air radar reflectivity in the atmospheric boundary layer, 2013 IEEE Antennas and Propagation Society International Symposium

Abstract from Muschinski, 2014 (MST14 in Sao Jose de Campos) : There have been numerous observational studies that show systematic differences between the mean vertical wind velocity and the mean Doppler velocity measured with a vertically pointing radar or sodar beam. Some of these biases are caused by deficiencies in the hardware or software of the radar or sodar system, but some biases are real, that is, they are of geophysical origin. One group of geophysical vertical-velocity biases results from non-zero covariances of reflectivity fluctuations and vertical-wind fluctuations within the resolution and during the dwell time. These covariances can be interpreted as reflectivity fluxes, which must not be confused with refractivity fluxes. Here, I present observations and computer simulations of reflectivity fluxes in the atmospheric boundary layer, and I discuss some of their theoretical and practical implications.   (I was not able to find a better reference)

To conclude, I find that this issue on the daytime downward bias of w is not well presented here. You should remove it, if you do not find a better way to include it in the discussion.

Line 227 and item 19 about (ISO, 1993): the website indicates:

'Only informative sections of standards are publicly available. To view the full content, you will need to purchase the standard by clicking on the "Buy" button.'

So thanks for the Appendix you added. Your remarks lines 678-685 are relevant. To illustrate Fig. A1, you could also add some comments about Fig 3 : near the surface, T is 20.2C (Table 3) and the relative humidity is close to 100%   (if I'm right). According to Fig. A1, the attenuation is close to its lowest value at 3 kHz whereas it is closed to a peaking value at 40 kHz.

And your remark on the altitude effect could come just after.

Line 246 : the suggestion from reviewer 2 is better than mine ($\rightarrow$ were obtained from an altitude).

Response to 30.3 : thanks a lot for your Fig. 1R.

This graph is interesting and may explain the σ value in Fig. 9a. You can see 'low' frequency eddies at 200m, peaking every 10 min (8h03, 8h13, 8h23,    , 8h43). ' low', when compared to the turbulence energy spectrum. This may be due to some physical pulses of turbulence (affecting w, or T or both). I agree with you that the time-height cross section would have been different if w had  been measured! But such a periodicity is probably not an accident.

It also seems to me that the minima at 200m, correspond to the cold events between 400 and 700 m.

Note: no clear periodicity appears in the data you provide for October the 15[th] (response 28) even after removing the diurnal tendency.

Anyhow, you are right. All of this is beyond the scope of your paper.

Response to 33.4 : usually, we compute vector averages of the wind  and calculate the standard deviation from the standard deviation of both components of the wind …. You can let it as it is, but tell it please.

Lines 300-301:

**along with the corresponding statistics for the data.**

> ➔  along with the corresponding statistics for the data and the received power for both PAA and acoustic speakers.

Line 310 : 'as shown in Fig. 5' can be removed since Pr is now in Fig. 6.

Lines 311-314 :

**gate is too close to the antenna, and factors including the recovery of the receiver and incomplete overlapping of the electromagnetic and acoustic beams due to the special separation between the antenna and speaker systems could lead to a significant gradient in the receiving power at this gate (Lataitis, 1992).**

> ➔ gate is too close to the antenna. Lataitis (1992) explained that factors including the recovery of the receiver and incomplete overlapping of the electromagnetic and acoustic beams due to the special separation between the antenna and speaker systems can lead to a significant gradient in the receiving power at this gate (Lataitis, 1992).

Line 317 :
**It is noteworthy that the most of the highest range gates** ➔ It is also noteworthy that the most of the highest range gates

Line 319 : my opinion is that the received power is the true limiting factor, when comparing the PAA to the acoustic speaker. I agree that the wind can also be a limiting factor, but it seems to me that the lower received power of the PAA is the major factor. However, your study of the wind effect is interesting and  I would not change anything.

Lines 338-343 :
RASS with respect to radiosonde were in good agreement with results reported in
previous studies (e.g., Görsdorf and Lehmann 2000), despite no correction for vertical velocity, which could have been partly because of the experiments being conducted on fine days with light wind and the application of a relatively long averaging time. In addition, removing the first gate data from the statistics may also contribute to the results.
> ➔ RASS with respect to radiosonde are in good agreement with results reported in previous studies (e.g., Görsdorf and Lehmann 2000), despite no correction for vertical velocity was done. This could be partly because the experiments were conducted on fine days with light wind and because of the application of a relatively long averaging time. In addition, removing the first gate data from the statistics may also have contributed to the good results.

Line 350 : **low peak power** ➔ lower received power (?)

Line 439 : **may also be reflected** ➔ is also reflected

Line 443 : see my remark for line 319. In Fig. 5c-d, 6b-c-d, the limiting factor is clearly the receiver threshold at -10dB…

Lines 536-537 :
sound pressure levels, the human body becomes warmer until death from hyperthermia has been estimated to occur at levels greater than 180 dB.
➔ sound pressure levels, the human body becomes warmer until death from hyperthermia. This has been estimated to occur at levels greater than 180 dB.

Line 592 : **frequency** ➔ its frequency
Anyhow, I do not find that this last sentence is very useful (lines 591-592). I would remove it.

Between lines 573 and 587, you should indicate in this paragraph, the beam width you finally selected.

---

## Author Response (AR2)

**Response to Reviewer 1 for the second review**

We are grateful to the reviewer for the very careful examination of our manuscript. We believe the revised manuscript is improved as a result.

1. Lines 45-48

**When using RASS techniques, one or more acoustic sources are co-located with an antenna, and the profiler provides the vertical profile of the speed at which the acoustic disturbance propagates vertically (Angevine et al., 1994) from the measurement of Doppler spectrum**

=> When using RASS techniques, one or more acoustic sources are co-located with an antenna, and the profiler (from the measurement of Doppler spectrum) provides the vertical profile of the speed at which the acoustic disturbance propagates vertically (Angevine et al., 1994).

I am still not satisfied with this: you can remove the words in brackets if you prefer.

**Response:** Done (Removed the text from Line 47).

2. Lines 53-59

**Thus, a vertical profile of the speed of sound can be converted to a profile of virtual temperature. The radial wind speed is considered in Eq. (1) because the neglect of the wind velocity along the beam is the largest source of error in RASS measurements (e.g., May et al., 1989; Angevine et al., 1994). However, we do not consider the radial wind speed in our experiments, because strong clutter sometimes contaminates the Doppler spectrum and masks the atmospheric echo in the vertical beam observation. This issue is addressed in later sections.**

=> Usually experimenters use the vertical beam to minimize the wind velocity correction. The neglect of w in Eq. (1) may be a large source of error in RASS measurements (e.g. May et al., 1989; Angevine et al., 1994). In spite of this, we were not able to consider w in this work because strong clutter sometimes contaminated the Doppler spectrum and masked the atmospheric echo in the vertical beam observation. This issue is addressed in later sections.

**Response:** We have revised the manuscript to correspond to this comment. However, we did not use $w$ as the reviewer suggested but retain "radial wind speed" based on comments from the 2$^{nd}$ reviewer.(Lines 53-57): "The radial wind speed is considered in Eq. (1) because the neglect of the wind velocity along the beam may be a largest source of error in RASS measurements (e.g., May et al., 1989; Angevine et al., 1994). However, we could not consider

the radial wind speed in our experiments, because strong clutter sometimes contaminated the Doppler spectrum and masked the atmospheric echo in the vertical beam observation.

3. Line 83 : are applied => were applied

**Response:** Done (Line 82).

4. Lines 136-138 :

**Indeed, the vertical velocity correction can decrease the accuracy of RASS in situations with calm wind and a lower reliability of vertical wind measurements (Görsdorf and Lehmann 2000).**

I do not approve this sentence, that cannot be used out of context to justify that you neglected w: there is not enough indication of the wind vertical velocities under your experimental conditions, even if it is early in the morning, under anticyclonic conditions and weak horizontal wind. No geographic indication is given to discard the possibility of sea breeze or mountain breeze for instance. I would then remove the sentence. The reader will easily understand that it is better not to make the velocity correction, when the error is likely to be larger than the correction.

**Response:** We have removed the sentence in the revised manuscript (Line 135).

5. Line 174 : 'flame' ????

Do you mean 'a vertical frame'? This could be a reason why you cannot provide measurements at low elevation (high elevation in this position) as required by Reviewer 2 (see her/his remark 2.3). As far as this remark is concerned, I cannot see how Fig. 2 has been revised. I'm wondering whether there would not be a confusion between the 'zenith angles' you indicate (0 to 40 deg) and the 'elevation angles' that Reviewer 2 mentions (0 to 40 deg ie 50 to 90 deg zenith angles). Anyhow I understood that your system is designed to work at high elevation ...

**Response:** This was a typo as the reviewer pointed out. We have revised the manuscript to respond to this comment (Line 171): "In order to measure the audible sound pressure level (SPL) pattern, we installed the PAA on a standing frame (Fig. 1b) for temporal use to radiate sound horizontally." We also revised Fig. 1a and its caption (Line 835) so that the standing frame appears more clearly. See Fig. 2R for detail.

6. Line 176 : **because a range of 10 m would be necessary to complete producing audible sound**

=> because a range of 10 m at least is necessary to produce audible sound ...(Prof. Kamakura, 2018, personal communication)

**Response:** We would retain this sentence because this was what Prof. Kamakura said to us. He was an expert of PAA speakers and has published a review paper on PAA as a co-author (Gan et al., 2012).

7. Line 178 : instead of the **personal communication,** could not this be simply explained by considerations of geometry?

**Response:** We think that the reviewer is considering something like the far field region (distance) of antenna determined by the length of the radio wave and the size of the antenna. However, the range to produce audible sound from ultrasound is determined not only by those geometric factors; rather, it is also determined by the nonlinear response of the air to the interaction between two collimated high-frequency sound beams, which might be a function of temperature, pressure, and/or humidity.

8. Line 205: **which may be preferable for the formation of the inversion layer** : I would not use 'preferable' since the quick variation of the inversion layer is more a drawback than a positive point. OK not to mention it now (and remove the highlighted text above). It is enough to mention this point and the resultant variability in the 1-hour results you show in the following.

**Response:** We have removed the text in the revised manuscript (Line 203) and added "with light winds" (Line 358) to read "since the operational radiosondes were launched in the morning of fine days with light winds, an inversion layer was frequently observed (Fig. 6)."

9. Lines 209-211:

9-1. It would not reduce the downward bias of the vertical velocity but the effect of this bias on the data dispersion. Neither Görsdorf and Lehmann 2000 nor Adachi et al., 2005 said that 1 hour averaging would reduce the bias, although they mentioned the downward bias of w and its effect on their results (10 min- averages and 30 min, respectively). Görsdorf and Lehmann 2000 suggested not to use the vertical velocity for long-term measurements and climatological investigations, to avoid the downward bias, but said that w should be taken into account for comparisons of individual profiles. However, shorter integration times (shorter than 10 minutes) would increase the systematic error in the w estimation. You perfectly know what Adachi and al. 2005 said...

9-2. **which could be attributable to insects or hydrometeors that are undetectable (Angevine, 1997)**

=> other factors could contribute : see for instance

Muschinski and Sullivan, 2013. Using large-eddy simulation to investigate intermittency fluxes of clear-air radar reflectivity in the atmospheric boundary layer, 2013 IEEE Antennas and Propagation Society International Symposium

Abstract from Muschinski, 2014 (MST14 in Sao Jose de Campos) : There have been numerous observational studies that show systematic differences between the mean vertical wind velocity and the mean Doppler velocity measured with a vertically pointing radar or sodar beam. Some of these biases are caused by deficiencies in the hardware or software of the radar or sodar system, but some biases are real, that is, they are of geophysical origin. One group of geophysical vertical-velocity biases results from non-zero covariances of reflectivity fluctuations and vertical-wind fluctuations within the resolution and during the dwell time. These covariances can be interpreted as reflectivity fluxes, which must not be confused with refractivity fluxes. Here, I present observations and computer simulations of reflectivity fluxes in the atmospheric boundary layer, and I discuss some of their theoretical and practical implications. (I was not able to find a better reference)

To conclude, I find that this issue on the daytime downward bias of w is not well presented here. You should remove it, if you do not find a better way to include it in the discussion.

**Response:**

9-1 and -2. We have removed the text in the revised manuscript (Line 206).

10. Your remarks lines 678-685 are relevant. To illustrate Fig. A1, you could also add some comments about Fig 3 : near the surface, T is 20.2C (Table 3) and the relative humidity is close to 100% (if I'm right). According to Fig. A1, the attenuation is close to its lowest value at 3 kHz whereas it is closed to a peaking value at 40 kHz.

And your remark on the altitude effect could come just after.

**Response:** We have revised the manuscript and added temperature and relative humidity to respond to this comment (Lines 678-679): the attenuation coefficient could increase with altitude for the former, while it decreases for the latter (e.g., Fig. 3, >1 km AGL, where $T = 20.2°C$ and $h_r = 76\%$ near the surface).

11. Line 246 : the suggestion from reviewer 2 is better than mine ( were obtained from an altitude).

**Response:** Done (Line 241).

12. Response to 33.4 : usually, we compute vector averages of the wind and calculate the standard deviation from the standard deviation of both components of the wind .... You can let it as it is, but tell it please.

**Response:** We have revised the caption for Table 3 to respond to this comment (Lines 830-831): "and mean wind speed aloft (20 — 1200 m AGL) with standard deviation. Means and standard deviations are not vector but scalar statistics.

13. Lines 300-301:

**along with the corresponding statistics for the data.**

=> along with the corresponding statistics for the data and the received power for both PAA and acoustic speakers.

**Response:** Done (Line 296).

14. Line 310 : 'as shown in Fig. 5' can be removed since Pr is now in Fig. 6.

**Response:** Done (Line 305).

15. Lines 311-314 :

**gate is too close to the antenna, and factors including the recovery of the receiver and incomplete overlapping of the electromagnetic and acoustic beams due to the special separation between the antenna and speaker systems could lead to a significant gradient in the receiving power at this gate (Lataitis, 1992).**

=> gate is too close to the antenna. Lataitis (1992) explained that factors including the recovery of the receiver and incomplete overlapping of the electromagnetic and acoustic beams due to the special separation between the antenna and speaker systems can lead to a significant gradient in the receiving power at this gate (Lataitis, 1992).

**Response:** We have revised our manuscript to respond to this comment (Line 306-310):" the first gate is too close to the antenna. In fact, Lataitis (1992) suggested that factors…speaker systems can lead to a significant gradient…"

16. Line 317 : **It is noteworthy that the most of the highest range gates** => It is also noteworthy that  most of the highest range gates

**Response:** Done (Line 313).

17. Lines 338-343 : RASS with respect to radiosonde were in good agreement with results reported in previous studies (e.g., Görsdorf and Lehmann 2000), despite no correction for vertical velocity, which could have been partly because of the experiments being conducted on fine days with light wind and the application of a relatively long averaging time. In addition, removing the first gate data from the statistics may also contribute to the results.

=> RASS with respect to radiosonde are in good agreement with results reported in previous studies (e.g., Görsdorf and Lehmann 2000), despite no correction for vertical velocity was done. This could be partly because the experiments were conducted on fine days with light wind and because of the application of a relatively long averaging time. In addition, removing the first gate data from the statistics may also have contributed to the good results.

**Response:** Done (Lines 334-339).

18. Line 350 : **low peak power** => lower received power (?)

**Response:** We intended to mention about the lower peak power transmitted from the PAA. We have modified our manuscript to make it clear (Line 346):" the low peak power mentioned previously (Fig. 2)."

19. Line 439 : **may also be reflected** => is also reflected

**Response:** Done (Line 435).

20. Line 443 : see my remark for line 319. In Fig. 5c-d, 6b-c-d, the limiting factor is clearly the receiver threshold at -10dB...

**Response:** We have modified our manuscript to correspond to this comment (Line 439):" the former can observe up to the highest range gate as the latter as long as the received power is more than about -10 dB.

21. Lines 536-537 : sound pressure levels, the human body becomes warmer until death from hyperthermia has been estimated to occur at levels greater than 180 dB.

=> sound pressure levels, the human body becomes warmer until death from hyperthermia. This has been estimated to occur at levels greater than 180 dB.

**Response:** Done (Lines 532-533)

22. Line 592 : **frequency** => its frequency

 Anyhow, I do not find that this last sentence is very useful (lines 591-592). I would remove it.

**Response:** We have removed the text in the revised manuscript (Line 587).

23. Between lines 573 and 587, you should indicate in this paragraph, the beam width you finally selected.

**Response:** We have revised our manuscript to respond to this comment (Line 576):" The sound wave was then steered windward with the default beam width (~5°)"